# Center-surround inhibition in expectation and its underlying computational and artificial neural network models

Ling Huang[1,2†‡], Shiqi Shen[1,2†], Yueling Sun[1,2†], Shipei Ou[2], Ru-Yuan Zhang[3], Floris P de Lange[4], Xilin Zhang[1,2*]

[1]Key Laboratory of Brain, Cognition and Education Sciences, Ministry of Education, South China Normal University, Guangzhou, China; [2]School of Psychology, Center for Studies of Psychological Application, and Guangdong Provincial Key Laboratory of Mental Health and Cognitive Science, South China Normal University, Guangzhou, China; [3]Brain Health Institute, National Center for Mental Disorders, Shanghai Mental Health Center, Shanghai Jiao Tong University School of Medicine and School of Psychology, Shanghai, China; [4]Donders Institute for Brain, Cognition and Behaviour, Radboud University, Nijmegen, Netherlands

*For correspondence:
xlzhang@m.scnu.edu.cn

†These authors contributed equally to this work

Present address: ‡Department of Psychology, The Ohio State University, Columbus, United States

## eLife Assessment

This is a methodologically rich manuscript that is **important** for revealing the center-surround inhibition profile of expectation in orientation space. The analyses are **compelling** in validating the critical role of predictive coding feedback. The findings provide novel insights into how expectation optimizes perception via enhancement and suppression.

**Abstract** Expectation is beneficial for adaptive behavior through quickly deducing plausible interpretations of information. The profile and underlying neural computations of this process, however, remain unclear. When participants expected a grating with a specific orientation, we found a center-surround inhibition profile in orientation space, which was independent from attentional modulations by task relevance. Using computational modeling, we showed that this center-surround inhibition could be reproduced by either a sharpening of tuning curves of expected orientation or a shift of tuning curves of unexpected orientations. Intriguingly, these two computations were further supported by orientation-adjustment and orientation-discrimination experiments. Finally, the ablation studies in convolutional neural networks revealed that predictive coding feedback played a critical role in the center-surround inhibition in expectation. Altogether, our study reveals for the first time that expectation results in both enhancement and suppression, optimizing plausible interpretations during perception by enhancing expected and attenuating similar but irrelevant and potentially interfering representations.

## Introduction

Human behavior is surprisingly efficient and adaptive. Although the everyday environment is brimming with noisy and ambiguous information, our cognitive system can quickly and adeptly deduce plausible interpretations of this information by combining it with prior expectations, ultimately facilitating flexible behavioral arises (*Bar, 2004*; *Bar, 2009*). However, the structured manner (the profile, in other words) regarding how expectation demarcates the anticipated target from various distractors

and underlying neural computations remains largely unclear. This issue is particularly important since such a profile is thought to closely reflect neural circuitry (*Teufel and Fletcher, 2020*; *Watabe-Uchida et al., 2017*), and therefore, offers us a unique opportunity to give insight into neural circuit level computations of expectation, thereby not only furthering our understanding of how it facilitates perception and behavior to adapt to changing environment, but also addressing a long-standing debate about its underlying neural mechanisms (*de Lange et al., 2018*; *Press et al., 2020*).

One of the central questions to this debate is about the processing of unexpected stimuli that are sufficiently novel or surprising. The sharpening models (also referred to as Bayesian theories) propose that expectations preferentially suppress neurons tuned toward the unexpected stimuli, resulting in a sharper and more selective population responses (*de Lange et al., 2018*; *Kok et al., 2012*; *Summerfield and de Lange, 2014*). This sharpening account of expectation is similar to the notion of neuronal resonance (*Lee and Mumford, 2003*) and has been supported by neurophysiological (*Bell et al., 2016*; *Fiser et al., 2016*; *Kaposvari et al., 2018*; *Meyer and Olson, 2011*; *Schwiedrzik and Freiwald, 2017*), electro-/magneto-encephalogram (*Aitken et al., 2020*; *Kok et al., 2017*; *Todorovic et al., 2011*; *Sedley et al., 2016*; *Wacongne et al., 2011*), and functional magnetic resonance imaging (fMRI; *Kok et al., 2012*; *Alink et al., 2010*; *Summerfield et al., 2008*; *Yon et al., 2018*) studies. Conversely, the cancelation models (also referred to as dampening theories) propose a dampening of neural responses reduces redundancy in the sensory system, through suppressing neurons tuned toward the expected stimulus. By canceling the expected information, the brain could highlight the processing and cognitive resources of unexpected information (*Press et al., 2020*; *Richter et al., 2022*). This theory has also drawn wide support from neurophysiological (*Meyer and Olson, 2011*; *Schwiedrzik and Freiwald, 2017*; *Kumar et al., 2017*) and brain imaging (*Blakemore et al., 1998*; *Blank and Davis, 2016*; *Han et al., 2019*; *Richter et al., 2018*) studies. Intriguingly, although these two models explaining how expectations render perception either veridical or informative are seemingly conflicting, both could be incorporated in the framework of predictive coding models (*Kok et al., 2012*; *Lee and Mumford, 2003*; *Feldman and Friston, 2010*; *Friston, 2005*; *Rao and Ballard, 1999*; *Summerfield and Koechlin, 2008*; *Yuille and Kersten, 2006*), which posits that the brain contains distinct neurons/units representing the best guess about the outside world (prediction units) and the discrepancy between these guesses and incoming sensory evidence (prediction error units). Several studies have proposed that the sharpening and cancelation accounts may occur in prediction and error neurons, respectively (*Press et al., 2020*; *Richter et al., 2022*; *Friston, 2005*). Within this framework, anticipating what is possible or probable in the forthcoming sensory environment can be cast as a process of hierarchical Bayesian inference, in which the prediction units are more strongly weighted towards the expected rather than unexpected stimuli, while at the same time the prediction error units are selectively biased to surprising inputs. Increased gain on these surprising inputs would lead to high-fidelity representations of unexpected stimuli across prediction units. Although, so far, there has been no direct evidence for the existence of these two neuron types, and it is unclear how these two mechanisms are reconciled from different neural populations, the predictive coding framework may provide the underlying computational basis for various potential profiles of expectation.

Here, given these two mechanisms making opposite predictions about how expectation changes the neural responses of unexpected stimuli, thereby displaying different profiles of expectation, we speculated that if expectation operates by the sharpening model with suppressing unexpected information, we should observe an inhibitory zone surrounding the focus of expectation, and its profile then should display as a center-surround inhibition (*Figure 1c*, left). If, however, expectation operates as suggested by the cancelation model with highlighting unexpected information, the inhibitory zone surrounding the focus of expectation should be eliminated, and the profile should instead display a monotonic gradient (*Figure 1c*, right). To adjudicate between these theoretical possibilities, we manipulated the distance between the expected and unexpected stimuli in feature space to measure the profile of expectation in two psychophysical experiments (orientation was task-relevant or task-irrelevant on the orientation and spatial frequency discrimination tasks, respectively, *Figure 1b*), both of which supported the sharpening account by showing a classical center-surround. inhibition profile in orientation space, with enhanced neural responses to the expected orientation and suppressed neural responses to orientations similar to the expected orientation relative to orientations more distinct from it (*Figure 2*). Second, using computational modeling, we showed that the behaviorally observed center-surround inhibition in expectation could be reproduced by either a sharpening of tuning curves

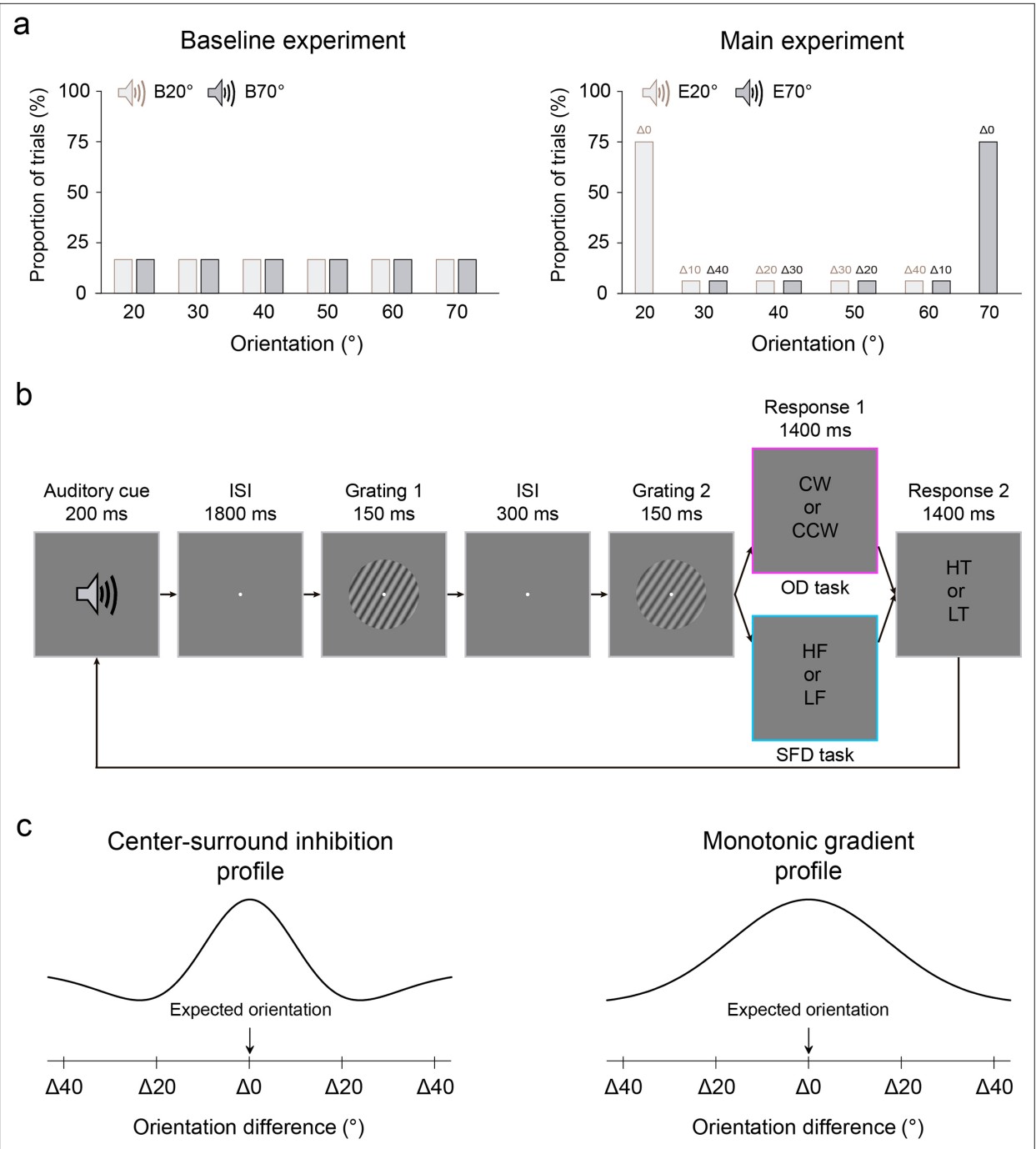

**Figure 1.** Stimuli and protocols of the profile experiment. (**a**) Left: the auditory cue, comprising either a low- or high-frequency tone, predicted the orientation of the first grating with equal validity in the baseline experiment. B20°: Baseline 20°; B70°: Baseline 70°. Right: in the main experiment, the low- or high-frequency tone predicted 20° or 70° (expected) orientation of the first grating with 75% validity. In the remaining 25% of trials, this orientation was chosen randomly and equally from four non-predicted orientations (30°, 40°, 50°, and 60°). There were two types of expected conditions: Expect 20° (E20°) and Expect 70° (E70°), and for both conditions, there were five possible distances in orientation space between the expected and test gratings, ranging from Δ0° through Δ40° with a step size of 10°. (**b**) In both baseline and main experiments, each trial began with an auditory cue, followed by an 1800 ms fixation interval. Then, two consecutive gratings were each presented for 150 ms and separated by a 300 ms blank interval. Participants were first asked to make a 2AFC judgment of either the orientation (clockwise or anticlockwise) or the spatial frequency (lower or higher) of the second grating relative to the first on orientation discrimination (OD, purple) and spatial frequency discrimination (SFD, blue) tasks, respectively. Then, participants were asked to make another 2AFC judgment on the tone of auditory cue, either low or high. CW: clockwise; CCW: counterclockwise; HF: higher frequency; LF: lower frequency; HT: high tone; LT: low tone. (**c**) Left: expectation operates by the sharpening model with suppressing unexpected information, under this configuration, the profile of expectation could display as a center-surround inhibition, with an inhibitory zone

*Figure 1 continued on next page*

*Figure 1 continued*

surrounding the focus of expectation. Right: expectation operates by the cancellation model with highlighting unexpected information. Under this configuration, the profile of expectation could display as a monotonic gradient, without the inhibitory zone.

of expected orientation (Tuning sharpening account) or a shift of tuning curves of unexpected orientations (Tuning shift account; *Figure 3a*). Third, these neural computations, consisting of both the tuning sharpening and tuning shift accounts, were further confirmed by orientation-adjustment (Figure 6) and orientation-discrimination (Figure 7) experiments. Finally, we found that a deep predictive coding neural network (DPCNN) exhibited a similar center-surround inhibition by expectation profile, both when it was trained to perform an orientation or a spatial frequency task. Most importantly, when we ablated predictive feedback, these center-surround inhibitions were eliminated in the DPCNN (Figure 8), strongly supporting the framework of predictive coding models in expectation (*Kok et al., 2012*; *Lee and Mumford, 2003*; *Feldman and Friston, 2010*; *Friston, 2005*; *Rao and Ballard, 1999*; *Summerfield and Koechlin, 2008*; *Yuille and Kersten, 2006*). Altogether, our study reveals for the first time that expectation generates an orientation-specific enhancement and suppression profile that optimizes plausible interpretations during visual perception by boosting expected and attenuating interfering sensory representations.

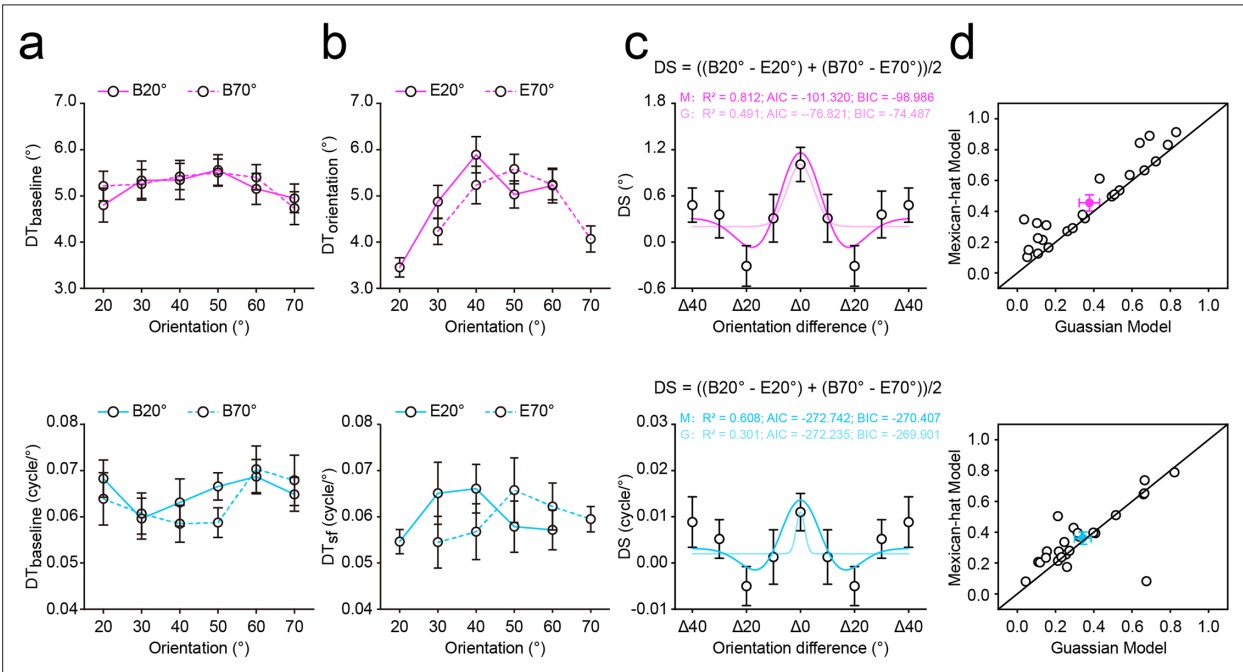

**Figure 2.** Results of the profile experiment. The discrimination thresholds of OD (top) and SFD (bottom) tasks during baseline (**a**) and main (**b**) experiments. In the baseline experiment, discrimination thresholds did not differ across orientations in either OD or SFD tasks, as confirmed by non-significant one-way repeated-measures ANOVAs (all p>0.18). B20°: Baseline 20°; B70°: Baseline 70°; E20°: Expect 20°; E70°: Expect 70°. (**c**) The averaged discrimination sensitivity (*DS*) of each distance on OD (top) and SFD (bottom) tasks, and the best fitting Gaussian and Mexican-hat functions to these DSs across distances. In both tasks, DS varied significantly across distances (OD: F(4,92) = 3.739, p=0.010, $\eta_p^2$=0.140; SFD: F(4,92) = 2.822, p=0.042, $\eta_p^2$=0.109), and *Post hoc* paired *t* tests revealed that, for both tasks, the *DSs* of Δ20° were significantly lower than those of both Δ0° and Δ40°, consistent with the classical center-surround inhibition profile. G, Gaussian model; M, Mexican-hat model. (**d**) $R^2$ of the best fitting Gaussian and Mexican-hat functions for individual participants in OD (top) and SFD (bottom) tasks. For both tasks, most dots located in the upper-left zone demonstrated that the Mexican-hat model was favored over the Gaussian model. Open symbols represent the data from each participant and filled colored dots represented the mean across participants. Error bars indicate 1 SEM calculated across participants (N = 24).

The online version of this article includes the following figure supplement(s) for figure 2:

**Figure supplement 1.** Accuracies of auditory tone reports in the profile experiment.

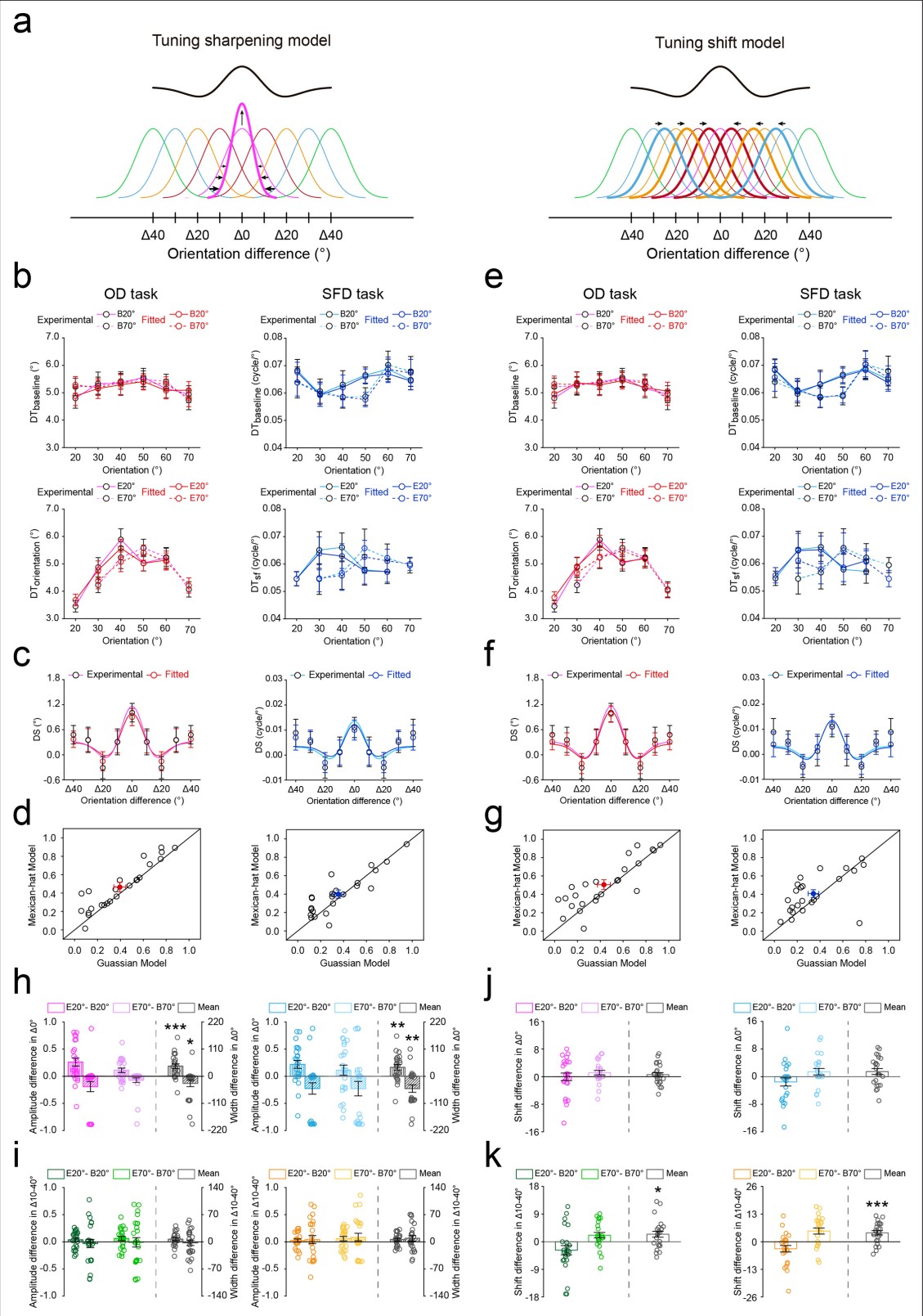

**Figure 3.** Results of computational modeling. (**a**) Illustration of the Tuning sharpening model (left) and the Tuning shift model (right). The Tuning sharpening model postulates that expectation sharpens the tuning of individual neurons (thick curves) towards the expected orientation, which results in a center-surround population response profile (black curve) centered at the expected orientation. The Tuning shift model postulates that expectation attracts the tuning of individual neurons (thick curves) from unexpected orientation towards the expected orientation, which also results in a center-

*Figure 3 continued on next page*

*Figure 3 continued*

surround population response profile. (**b**) The fitted discrimination thresholds on OD (left) and SFD (right) tasks in the baseline (top) and main (bottom) experiments. (**c**) The averaged DSs using Tuning sharpening model on OD (left) and SFD (right) tasks. (**d**) $R^2$ of the best fitting Gaussian and Mexican-hat functions for individual participants based on the fitted DSs using Tuning sharpening model on OD (left) and SFD (right) tasks. Open symbols represent the data from each participant and filled colored dots represented the mean across participants. (**e–g**) The results from the Tuning shift model, see caption for (**b–d**) for a description of each type of graph. The amplitude *A* (vertical stripes) and width *σ* (diagonal stripes) differences between the baseline and main experiments using Tuning sharpening model in Δ0° (**h**) and Δ10°-Δ40° (**i**) conditions, on OD (left) and SFD (right) tasks. The location *x0* differences between the baseline and main experiments using Tuning shift model in Δ0° (**j**) and Δ10°-Δ40° (**k**) conditions, on OD (left) and SFD (right) tasks. Statistical comparisons were performed using t-tests against zero. Open symbols represent the data from each participant, and error bars indicate 1 SEM calculated across participants (N = 24). B20°: Baseline 20°; B70°: Baseline 70°; E20°: Expect 20°; E70°: Expect 70° (*p<0.05; **p<0.005; ***p<0.001).

The online version of this article includes the following figure supplement(s) for figure 3:

**Figure supplement 1.** The calculated parameters of the population responses on OD and SFD tasks.

**Figure supplement 2.** Results of combined computational modeling.

# Results

## Profile experiment

The profile experiment consisted of a baseline and main experiment, with the baseline experiment always preceding the main experiment. The two experiments were the same, except for the probability relationship between the auditory cue and the orientation (20°, 30°, 40°, 50°, 60°, and 70°) of the first grating. For the baseline experiment, the auditory cue, comprising either a low- (240 Hz) or high- (540 Hz) frequency tone, predicted the orientation of the first grating with equal validity (16.67%, *Figure 1a*, left). In the main experiment, this low- or high-frequency tone auditory cue predicted the orientation (20° or 70°) of the first grating with 75% validity. In the remaining 25% of trials, this orientation was chosen randomly and equally from four non-predicted orientations (30°, 40°, 50°, and 60°, *Figure 1a*, right). Thus, for each participant, there were two types of expected conditions: Expect 20° and Expect 70°, and for both conditions, there were five possible distances in orientation space between the expected and test gratings, ranging from Δ0° through Δ40° with a step size of 10°. Note that the matches between the tone (low- or high-frequency) of auditory cue and the expected orientation (20° or 70°) of the first grating were flipped across participants, and the order was also counterbalanced across participants. Moreover, for each participant, although the tone of auditory cue could not predict 20° or 70° orientation in the baseline experiment, the trials in the baseline experiment with the same tone that was matched with 20° or 70° orientation in the main experiment were defined as Baseline 20° (i.e. the baseline of Expect 20°) and Baseline 70° (i.e. the baseline of Expect 70°) conditions, respectively.

Both the baseline and main experiments consisted of two tasks: the orientation discrimination (OD) task and spatial frequency discrimination (SFD) task. With the two tasks occurring on different days, the order of the two tasks was counterbalanced across participants. Differently, the baseline experiment consisted of four blocks (two for OD task and the other two for SFD task), and each block had two QUEST staircases (*Watson and Pelli, 1983*) for each of six orientations (20°, 30°, 40°, 50°, 60°, and 70°). The main experiment consisted of 2 blocks (one for OD task and the other one for SFD task), and each block had 24 QUEST staircases for the expected orientations (20° and 70°) and 2 QUEST staircases for each of unexpected orientations (30°, 40°, 50°, and 60°). Each QUEST staircase comprised 40 trials and each trial began with an auditory cue, followed by a fixation interval. Then, two gratings were presented sequentially, and participants were asked to make a two-alternative forced-choice (2AFC) judgment of either the orientation (clockwise or anticlockwise, where orientation was task-relevant) or the spatial frequency (lower or higher, where orientation was task-irrelevant) of the second grating relative to the first, on the OD and SFD tasks, respectively (*Figure 1b*). The second grating differed trial by trial from the first in either orientation (Δθ°) or spatial frequency (Δλ cycles/°) on the OD and SFD tasks, respectively. The QUEST staircase was used to control the varied Δθ° or Δλ cycles/° adaptively for estimating participants' discrimination thresholds (75% correct). At the end of each trial, participants needed to report the tone (either low or high) of the auditory cue. For either OD or SFD tasks, there was no significant difference in accuracy of this reporting across different conditions in either baseline or main experiments (*Figure 2—figure supplement 1*).

In the baseline experiment, participants' mean discrimination thresholds in Baseline 20° and Baseline 70° conditions were submitted to a one-way repeated-measures ANOVA with orientation (20°, 30°, 40°, 50°, 60°, and 70°) as a within-participants factor. Results showed that the main effect of orientation was not significant in either OD (Baseline 20°: F(5,115) = 0.955, p=0.431, $\eta_p2$ = 0.040; Baseline 70°: F(5,115) = 1.314, p=0.274, $\eta_p2$ = 0.054) or SFD (Baseline 20°: F(5,115) = 1.163, p=0.331, $\eta_p2$ = 0.048; Baseline 70°: F(5,115) = 1.593, p=0.184, $\eta_p2$ = 0.065) tasks (*Figure 2a*), indicating that there was no significant difference in participant performance among six orientations. In other words, the tone of auditory cue in the baseline experiment was uninformative about the orientation of gratings. On both OD and SFD tasks and both two expected conditions (Expect 20° and Expect 70°), for each distance (Δ0°-Δ40°), we computed a discrimination sensitivity (*DS*) to quantify how much the discrimination threshold (*DT*) changed between baseline (*DT baseline*) and main (*DT main*) experiments: *DS = DT baseline - DT main*. Because the *DS* from Expect 20° and Expect 70° showed a similar pattern, they were pooled together for further analysis (unless otherwise stated, we present average data from two expected conditions). The averaged *DS*s were submitted to a one-way repeated-measures ANOVA with the distance (Δ0°-Δ40°) as a within-participants factor. Results showed that the main effect of distance was significant in both OD (F(4,92) = 3.739, p=0.010, $\eta_p2$ = 0.140, *Figure 2c*, top) and SFD (F(4,92) = 2.822, p=0.042, $\eta_p2$ = 0.109, *Figure 2c*, bottom) tasks. To directly address the potential inhibitory zone surrounding the focus of expectation, we compared the *DS*s between Δ20° and Δ0°, and between Δ20° and Δ40° on each task. Post hoc paired *t* tests revealed that, for both tasks, the *DS*s of Δ20° were significantly lower than those of both Δ0° (OD task: t(23) = –4.263, p<0.001, Cohen's *d*=0.870; SFD task: t(23) = –4.679, p<0.001, Cohen's *d*=0.955) and Δ40° (OD task: t(23) = –2.214, p=0.037, Cohen's *d*=0.452; SFD task: t(23) = –2.694, p=0.013, Cohen's *d*=0.550), indicating a classical center-surround inhibition in expectation with the enhanced discriminability to the expected orientation (Δ0°) and decreased discriminability to orientations (Δ20°) similar to the expected orientation relative to orientations (Δ40°) more distinct from it. Intriguingly, this center-surround inhibition profile of expectation was independent of attentional modulations by task relevance of the orientation.

Subsequently, to further assess the shape of this expectation pattern, we fitted a monotonic model and a non-monotonic model to the average *DS*s across distances (Δ0°-Δ40°) on both OD and SFD tasks. The monotonic and nonmonotonic models were implemented as the Gaussian and Mexican-hat functions, respectively (*Shen et al., 2024*; *Wang et al., 2021*). To compare these two models to our data, we first computed the Akaike information criterion (AIC; *Akaike, 1973*) and Bayesian information criterion (BIC; *Schwarz, 1978*) with the assumption of a normal error distribution. Then, we calculated the Likelihood ratio (LR) and Bayes factor (BF) of the Mexican-hat model over the Gaussian model based on AIC (*Burnham and Anderson, 2002*) and BIC (*Wagenmakers, 2007*) approximation, respectively. Results showed that, in both tasks, the LR/BFs were larger than 1 (OD task: LR/BF = $2.088 \times 10^5$; SFD task: LR/BF = 1.288) and therefore strongly favored the Mexican-hat model over the Gaussian model (*Figure 2c*). Notably, we also conducted similar model comparisons for each participant's data and found that the Mexican-hat model was favored over the Gaussian model in 23 and 17 of 24 participants, for OD and SFD tasks, respectively (*Figure 2d*). Together, these results constituted strong evidence for the center-surround inhibition profile of expectation and further indicated its independence of attentional modulations by task relevance of the orientation.

## Computational models of the center-surround inhibition in expectation

Our results demonstrated the classical center-surround inhibition profile in expectation, yet it remains unclear what type of neural computations could account for this profile. We proposed that this profile could be explained by either Tuning sharpening (*Figure 3a*, left) or Tuning shift (*Figure 3a*, right) models. The Tuning sharpening model postulates that expectation sharpens the tuning of individual neurons (thick curves) of the expected orientation, which results in a center-surround population response profile (black curve) centered at the expected orientation. The Tuning shift model postulates that expectation attracts the tuning of individual neurons (thick curves) from unexpected orientations towards the expected orientation, which also results in a center-surround population response profile. Note that, in our study, the shift towards 20° was (arbitrarily) considered to be the negative value, whereas the shift towards 70° was thus the positive value, and unless otherwise stated, we present the average shift, that is *mean shift = (shift towards 70° - shift towards 20°)/2*, across conditions hereafter. For both OD and SFD tasks, to compare these two models, we fitted both the Tuning sharpening and

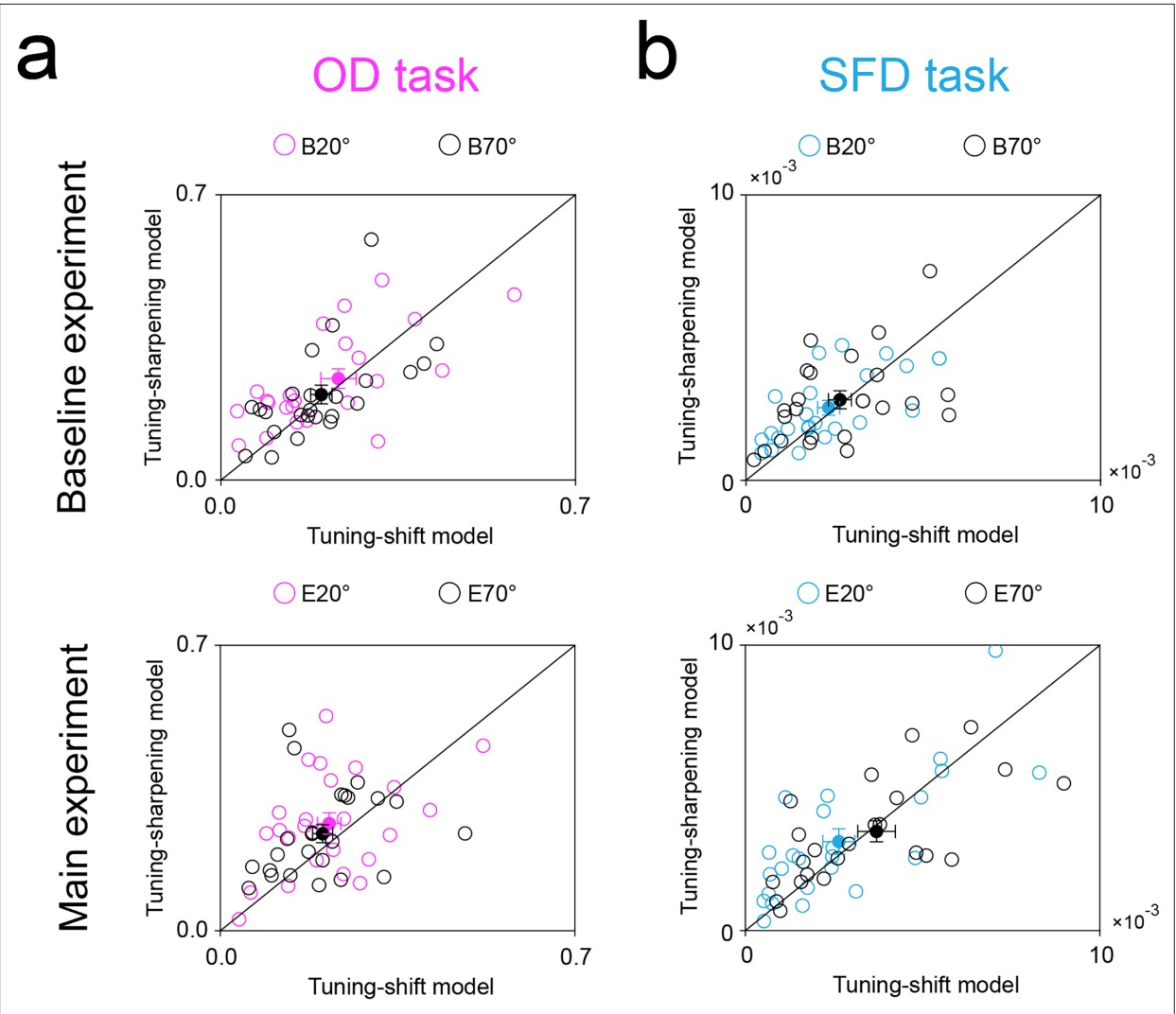

**Figure 4.** RMSDs of Tuning sharpening and Tuning shift models. RMSDs of Tuning sharpening and Tuning shift models during the baseline (top) and main (bottom) experiments, on OD (**a**) and SFD (**b**) tasks. B20°: Baseline 20°; B70°: Baseline 70°; E20°: Expect 20°; E70°: Expect 70°. Open symbols represent the data from each participant and filled colored dots represented the mean across participants. Error bars indicate 1 SEM calculated across participants (N = 24).

Tuning shift models (sum of idealized channel tuning functions) to the population response profiles (the smooth negative values of discrimination thresholds) during baseline and main experiments, and measured their root mean squared deviation (RMSD) metric (*Pitt et al., 2002*). RMSD takes the number of model parameters into account, and a smaller RMSD indicates better model fitness. Our results showed that both models exhibited robust fits to our data (*Figure 3b and e*), as indicated by high $R^2$ values and comparably low RMSDs in both OD (*Figure 4a*) and SFD (*Figure 4b*) tasks. Similarly, we computed a discrimination sensitivity (*DS*) to quantify how much the fitted discrimination threshold (*FDT*) changed between baseline ($FDT_{baseline}$) and main ($FDT_{main}$) experiments: DS = $FDT_{baseline}$ - $FDT_{main}$. For both models, a similar center-surround inhibition profile of the *DS* was found on both OD (*Figure 3c*) and SFD (*Figure 3f*) tasks. Further model comparisons for each participant's data confirmed that the Mexican-hat model was favored over the Gaussian model on both OD (22 and 19 of 24 participants for Tuning sharpening and Tuning shift models, respectively, *Figure 3d*) and SFD (14 and 17 of 24 participants for Tuning sharpening and Tuning shift models, respectively, *Figure 3g*) tasks. These results imply that the center-surround inhibition in expectation could be reproduced by either Tuning sharpening or Tuning shift models.

For each model and each task, to directly compare the tuning curve changes of both the expected (Δ0°) and unexpected orientations (Δ10°-Δ40°) with and without expectation, we calculated the parameters changes of tuning functions (amplitude $A$, location $x0$, and width $\sigma$) for hypothesized channels between baseline and main experiments (*Figure 3—figure supplement 1*). For the Tuning sharpening model, the tuning width of each channel's tuning function is parameterized by $\sigma$, while all tuning functions are evenly distributed with 10° spacing on the $x$-axis and the areas under the curves (response energy) are identical. Conversely, for the Tuning shift model, the location of each channel's tuning function is parameterized by $x0$, while they all share the same tuning amplitude and width. For both models, parameters were varied to obtain the minimal sum of squared errors between the population response profile and the model prediction, which is the sum of all channels' tuning responses. For the expected orientation (Δ0°) of Tuning sharpening model, results showed that the amplitude change was significantly higher than zero on both OD ($t(23) = 4.198$, $p<0.001$, Cohen's $d=0.857$) and SFD ($t(23) = 3.247$, $p=0.004$, Cohen's $d=0.663$) tasks (*Figure 3h*, vertical stripes); the width change was significantly lower than zero on both OD ($t(23) = -2.235$, $p=0.035$, Cohen's $d=0.456$) and SFD ($t(23) = -3.313$, $p=0.003$, Cohen's $d=0.676$) tasks (*Figure 3h*, diagonal stripes). For unexpected orientations (Δ10°-Δ40°), however, the amplitude and width changes were not significant with zero on either OD (amplitude change: $t(23) = 1.948$, $p=0.064$, Cohen's $d=0.397$; width change: $t(23) = -0.412$, $p=0.684$, Cohen's $d=0.084$) or SFD (amplitude change: $t(23) = 1.708$, $p=0.101$, Cohen's $d=0.349$; width change: $t(23) = 1.273$, $p=0.216$, Cohen's $d=0.260$) tasks (*Figure 3i*). For the Tuning shift model, results showed that the location shift was significantly different than zero for unexpected orientations (Δ10°-Δ40°), OD task: $t(23) = 2.547$, $p=0.018$, Cohen's $d=0.520$; SFD task: $t(23) = 4.099$, $p<0.001$, Cohen's $d=0.837$ (*Figure 3k*), but not for the expected orientation Δ0°, OD task: $t(23) = 0.993$, $p=0.331$, Cohen's $d=0.203$; SFD task: $t(23) = 1.750$, $p=0.093$, Cohen's $d=0.357$ (*Figure 3j*). These results further confirm the Tuning sharpening and Tuning shift computations for the center-surround inhibition in expectation.

In addition, across participants, we further used the non-parametric Wilcoxon signed-rank test to compare both the $R^2$ and RMSD between two models for Baseline 20°, Baseline 70°, Expect 20°, and Expect 70° conditions during each task. Results showed that there was no significant difference between two models in Baseline 20° (OD task: $R^2$: $z=1.372$, $p=0.170$, effect size: $r=0.280$; RMSD: $z=1.200$, $p=0.230$, effect size: $r=0.245$; SFD task: $R^2$: $z=0.857$, $p=0.391$, effect size: $r=0.175$; RMSD: $z=0.829$, $p=0.407$, effect size: $r=0.169$), Baseline 70° (OD task: $R^2$: $z=0.371$, $p=0.710$, effect size: $r=0.076$; RMSD: $z=0.029$, $p=0.977$, effect size: $r=0.006$; SFD task: $R^2$: $z=1.657$, $p=0.097$, effect size: $r=0.338$; RMSD: $z=0.686$, $p=0.493$, effect size: $r=0.140$), Expect 20° (OD task: $R^2$: $z=0.686$, $p=0.493$, effect size: $r=0.140$; RMSD: $z=1.600$, $p=0.110$, effect size: $r=0.327$; SFD task: $R^2$: $z=1.257$, $p=0.209$, effect size: $r=0.257$; RMSD: $z=1.600$, $p=0.110$, effect size: $r=0.327$), or Expect 70° (OD task: $R^2$: $z=1.486$, $p=0.137$, effect size: $r=0.303$; RMSD: $z=1.686$, $p=0.092$, effect size: $r=0.344$; SFD task: $R^2$: $z=0.514$, $p=0.607$, effect size: $r=0.105$; RMSD: $z=0.143$, $p=0.886$, effect size: $r=0.029$) conditions (*Figure 4*). These results further imply that Tuning sharpening and Tuning shift models may jointly contribute to center-surround inhibition in expectation.

To further examine whether both mechanisms jointly explain the observed center–surround inhibition under expectation, we also tested a combined model that incorporates tuning sharpening for the expected orientations (Δ0°) and tuning shift for the unexpected orientations (Δ10°-Δ40°) with and without expectation. This model successfully captured the sharpening of the expected-orientation channel and the shift of the unexpected-orientation channels (*Figure 3—figure supplement 2*), providing further evidence that tuning sharpening and tuning shift jointly contribute to center–surround inhibition in expectation.

## Orientation adjustment experiment

Experimentally, to further explore the co-existence of both Tuning sharpening and Tuning shift computations in center-surround inhibition profile of expectation, participants were asked to perform a classic orientation adjustment experiment. Unlike the profile experiment (discrimination tasks), the adjustment experiment provides a direct, trial-by-trial measure of participants' perceived orientation, capturing the full distribution of responses. This enables the construction of orientation-specific tuning curves, allowing us to detect both tuning sharpening and tuning shifts, thereby offering a more nuanced understanding of the computational mechanisms underlying expectation. The protocol of orientation adjustment experiment was similar to that of the profile experiment, except for two

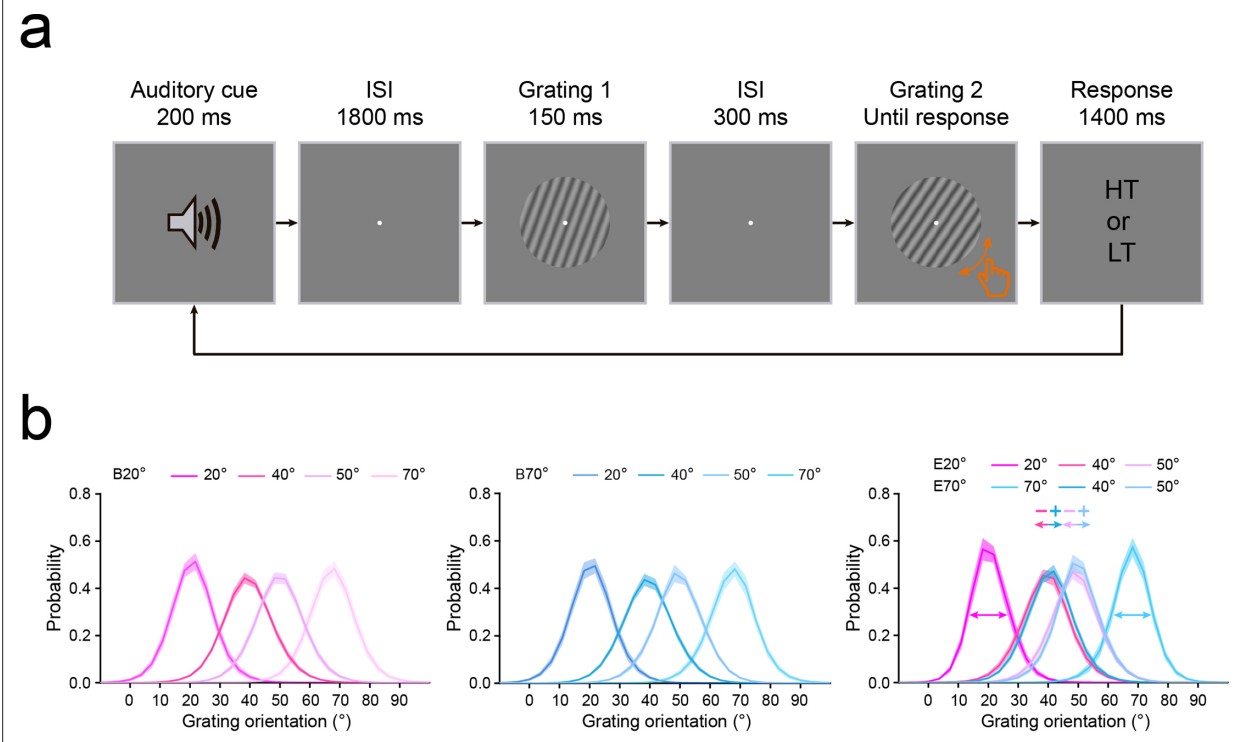

**Figure 5.** Protocol and error distributions of the orientation adjustment experiment. (**a**) The protocol of orientation adjustment experiment was similar to that of the profile experiment, except for two aspects. First, there were four possible (20°, 40°, 50°, and 70°) orientations for the first grating: 20°/70° (Δ0° deviated from the expected orientation) and 40°/50° (Δ20°/Δ30° deviated from the expected orientation). Second, in both baseline and main experiments, the second grating was set as a random orientation within the range of 0° to 90°, and participants were required to rotate the orientation of the second grating to match the first. HT: high tone; LT: low tone. (**b**) Three-component mixture model to the adjusted errors from baseline (left) and main (middle) experiments. In the current study, the shift towards 20° was (arbitrarily) considered to be the negative value ('-'), whereas the shift towards 70° was thus the positive value ('+'). The mean shift was calculated as: *mean shift = (shift towards 70° - shift towards 20°)/2.* The shaded error bars indicate 1 SEM calculated across participants (N = 20). B20°: Baseline 20°; B70°: Baseline 70°; E20°: Expect 20°; E70°: Expect 70°.

The online version of this article includes the following figure supplement(s) for figure 5:

**Figure supplement 1.** Accuracies of auditory tone report in orientation adjustment experiments.

aspects. First, there were four possible (20°, 40°, 50°, and 70°) orientations for the first grating: 20°/70° (Δ0° deviated from the expected orientation) and 40°/50° (Δ20°/Δ30° deviated from the expected orientation). Second, in both baseline and main experiments, the second grating was set as a random orientation within the range of 0° to 90°, and participants were required to rotate the orientation of the second grating to match the first (*Figure 5a*). Similar to the profile experiment, no significant difference was found in tone report accuracies across distances (*Figure 5—figure supplement 1*). For both expected (Δ0°) and unexpected (Δ20°/Δ30°) orientations, we calculated the adjusted orientation difference between the baseline and main experiments. Results showed the adjusted difference was significantly higher than zero for unexpected orientations (0.735±0.308: t(19) = 2.387, p=0.028, Cohen's d=0.534, *Figure 6a*, right), but not for the expected orientation (0.143±0.523: t(19) = 0.274, p=0.787, Cohen's d=0.061, *Figure 6b*, left), suggesting a significant bias in the unexpected orientation representation towards the expected orientation.

Subsequently, we employed a three-component mixture model (*Suchow et al., 2013*; *Zhang and Luck, 2008*) to the adjusted errors from both baseline and main experiments (*Figure 5b*). This allowed us to estimate representation precision, including the mean shift (*mu*) and standard deviation (*s.d.*) of the von Mises distribution (positive values indicating rightward shift and higher values indicating lower precision, respectively), along with assessing the probability of stimulus guessing (*g*). Using the difference between the main and baseline experiments (*Figure 6—figure supplement 1*), we also found that the orientation representation significantly shifted for unexpected orientations (0.752±0.303: t(19) = 2.481, p=0.023, Cohen's d=0.555, *Figure 6b*, right), but not for the expected orientation

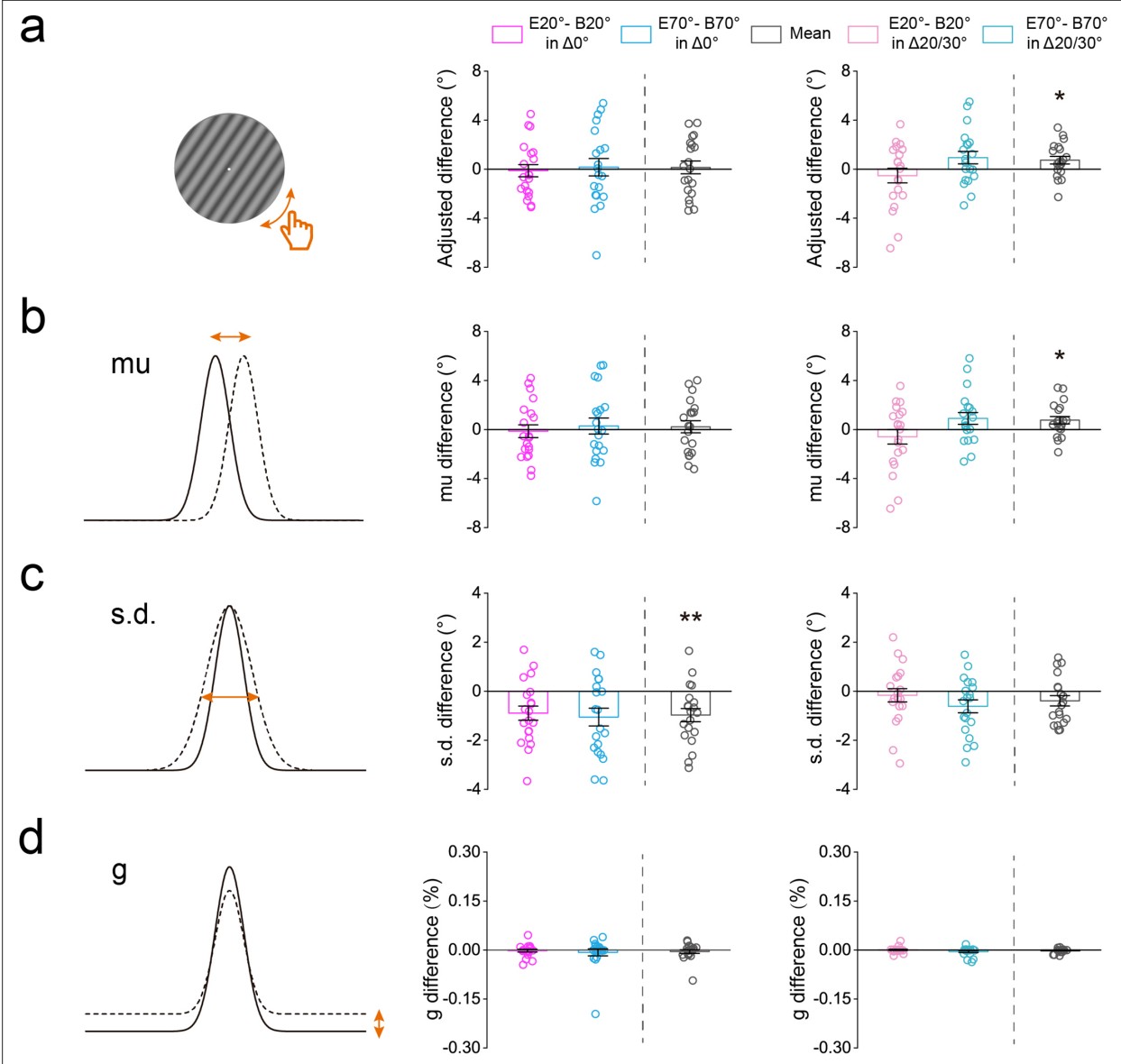

**Figure 6.** Results of the orientation adjustment experiment. (**a**) The adjusted orientation difference between the baseline and main experiments in both expected (20°/70°, i.e. Δ0°, middle) and unexpected (40°/50°, i.e. Δ20°/Δ30°, right) conditions. B20°: Baseline 20°; B70°: Baseline 70°; E20°: Expect 20°; E70°: Expect 70°. (**b–d**) The parameter estimates difference between the baseline and main experiments in both expected (Δ0°, middle) and unexpected (Δ20°/Δ30°, right) orientations. The parameter estimates were obtained by fitting a three-component mixture model to adjusted errors in different conditions. (**b**) *mu* reflects the response distribution shift away from the presented grating orientation. (**c**) *s.d.* reflects precision of responses (with higher values indicating worse precision). (**d**) *g* estimates the probability that the participant produced a random response (i.e. the guess). Statistical comparisons were performed using t-tests against zero. Open symbols represent the data from each participant and error bars indicate 1 SEM calculated across participants (N = 20; *p<0.05; **p<0.005; ***p<0.001).

The online version of this article includes the following figure supplement(s) for figure 6:

**Figure supplement 1.** Adjusted errors of orientation adjustment experiments and their parameter estimates with the three-component mixture model.

(0.214±0.493: t(19) = 0.434, p=0.669, Cohen's d=0.097, *Figure 6b*, left). Conversely, participants exhibited higher orientation representation precision than baseline in the expected orientation (–0.973±0.271: t(19) = –3.597, p=0.002, Cohen's d=0.804, *Figure 6c*, left), but not in unexpected orientations (–0.390±0.212: t(19) = –1.837, p=0.082, Cohen's d=0.411, *Figure 6c*, right). Finally, we found no significant difference with zero in the rate of guessing in either expected (–0.0047±0.0554: t(19) = –0.849, p=0.406, Cohen's d=0.190, *Figure 6d*, left) or unexpected (–0.0023±0.0148: t(19) =

–1.551, p=0.138, Cohen's d=0.347, *Figure 6d*, right) orientations. These results provide converging evidence supporting our hypothesis that both Tuning sharpening and Tuning shift contribute to the center-surround inhibition profile of expectation.

## Orientation discrimination experiment

Note that behavioral benefits in our orientation adjustment task could be due to improvements in either perceptual or decisional processes, as the expectation cue held information about both the most likely stimulus and the most likely correct response (*de Lange et al., 2018*; *Kok et al., 2012*; *Aitken et al., 2020*; *Kok et al., 2017*). To remove this link between stimulus and response expectations and thereby avoid potential response biases induced by the cue, we designed an additional orientation discrimination experiment. The protocol of this orientation discrimination experiment was very similar to that of the orientation adjustment experiment, except for two aspects (*Figure 7a*). First, there were three possible (20°, 45°, and 70°) orientations for the first grating: 20°/70° (Δ0° deviated from the expected orientation) and 45° (Δ25° deviated from the expected orientation). Second, in both baseline and main experiments, the second grating was 1°, 3°, 5°, 7°, and 9° deviated from the first grating, either clockwise or counterclockwise. Participants were asked to make a 2AFC judgment of the orientation of the second grating relative to the first, either clockwise or anticlockwise. *Figure 7b and c* show the psychometric functions for each condition. We plotted the percentage of trials in which participants indicated the orientation of the second grating that was anticlockwise or clockwise to the first for 20° (Baseline 20° and Expect 20°) and 70° (Baseline 70° and Expect 70°) conditions, respectively, as a function of the actual orientation difference between the two gratings. For each participant and each condition, the psychometric values at 10 orientation differences were fitted to a cumulative Gaussian function, and we interpolated the data to find the slope (orientation uncertainty) and PSE (point of subjective equality, which is the shift here) as an index for Tuning sharpening and Tuning shift models, respectively.

Similar to the orientation adjustment experiment, no significant difference was found in tone report accuracies across distances (*Figure 7—figure supplement 1*). For both expected (Δ0°) and unexpected (Δ25°) orientations, we calculated the slope and shift difference between the baseline and main experiments. Results showed the slope difference was significantly higher than zero for expected orientations (0.0250±0.0075: t(17) = 3.324, p=0.004, Cohen's d=0.627, *Figure 7d*, top), but not for the unexpected orientation (–0.0204±0.0113: t(17) = –1.812, p=0.088, Cohen's d=0.402, *Figure 7d*, bottom). Conversely, the shift difference was significantly lower than zero for the unexpected orientation (–0.696±0.287: t(17) = –2.423, p=0.027, Cohen's d=0.507, *Figure 7e*, bottom), but not for expected orientations (0.449±0.509: t(17) = 0.881, p=0.391, Cohen's d=0.452, *Figure 7e*, top). These results indicated that the expectation not only sharpened the tuning curves of neurons for the expected orientation but also attracted the tuning curves of neurons for unexpected orientations, further confirming Tuning sharpening and Tuning shift models, respectively, in the center-surround inhibition of expectation.

## Artificial neural networks for the center-surround inhibition in expectation

Finally, we trained a deep predictive coding neural network (DPCNN), modified from Predify (*Choksi and Mozafari, 2021*; *Pang et al., 2021*) to perform both the OD and SFD tasks. For both tasks, the DPCNN consisted of six feedforward encoding layers (*e1-e6*), five generative feedback decoding layers (*d1-d5*), and three fully connected (fc) layers (*Figure 8a*). The reconstruction error (*E1-E5*) is computed and used for the proposed predictive coding updates (*Rao and Ballard, 1999*), denoted by *P.C.* loops. Note that the updating is only applied to *e1-e5*, and for the last layer *e6*, there is no feedback. Before the layer *e0*, we obtained the pixel difference between the target and reference images, which was then superimposed on the channels of the reference image. This superimposed image was set as the input of the network and remained constant over timesteps. Besides, we used a feedforward encoding layer (i.e. *e0*) to match the number of the channels between superimposed feature maps and the pre-trained DPCNN. During the training, the last layer of the network was trained to capture the difference between the target and reference and finally obtain the classification by softmax, to model decision making in our 2AFC paradigm (*Figure 1c*), in which participants were asked to make a 2AFC judgment of the orientation (either clockwise or anticlockwise) or the spatial frequency (either lower or higher) of the second grating (target) relative to the first (reference) in OD

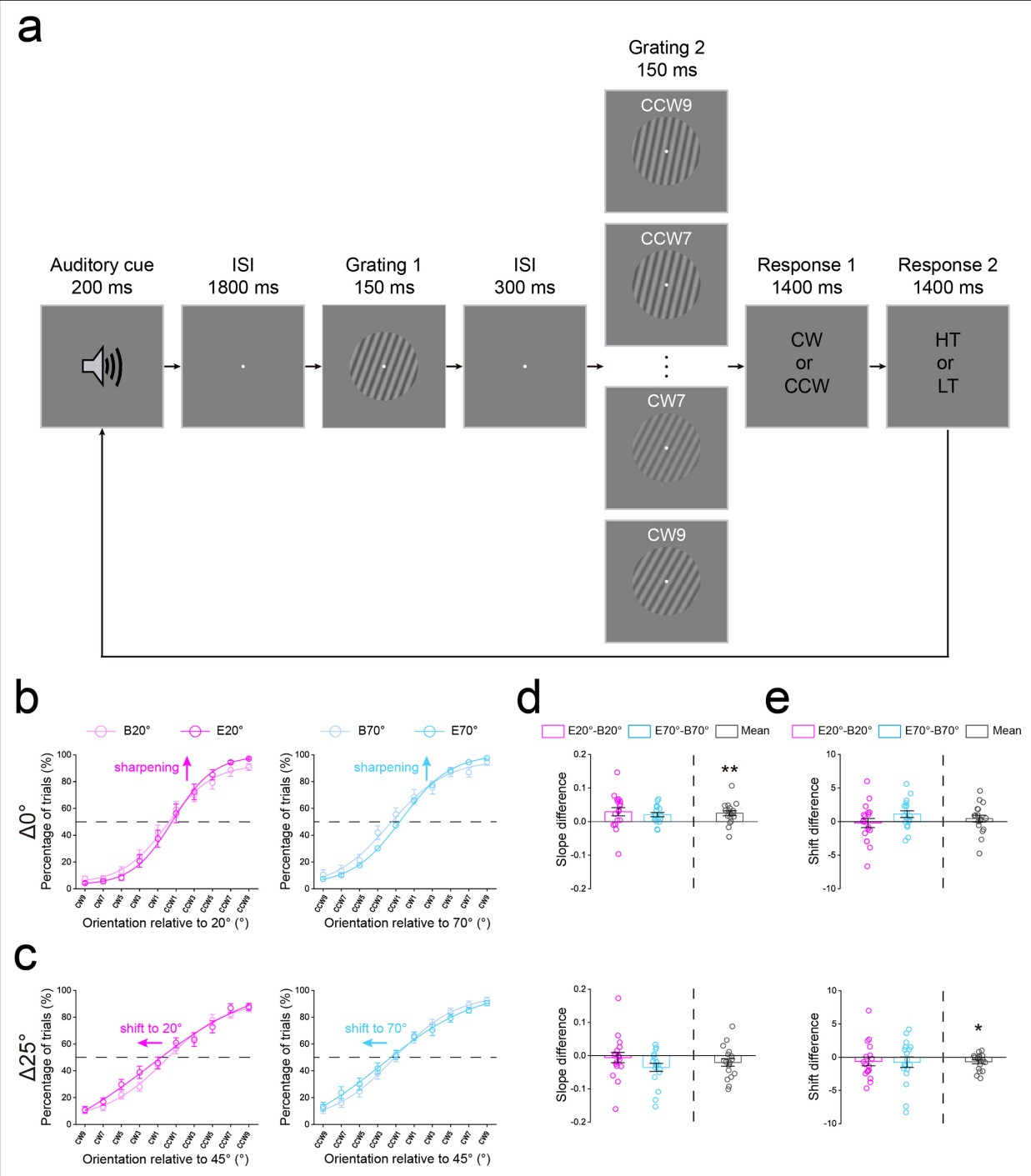

**Figure 7.** Protocol and results of the orientation discrimination experiment. (**a**) The protocol of orientation discrimination experiment was similar to that of the orientation adjustment experiment, except for two aspects. First, there were three possible (20°, 45°, and 70°) orientations for the first grating: 20°/70° (Δ0° deviated from the expected orientation) and 45° (Δ25° deviated from the expected orientation). Second, in both baseline and main experiments, the second grating was 1°, 3°, 5°, 7°, and 9° deviated from the first grating, either clockwise (CW) or counterclockwise (CCW). Participants were asked to make a 2AFC judgment of the orientation of the second grating relative to the first, either clockwise or anticlockwise. HT: high tone; LT: low tone. Psychometric functions showing orientation judgements in each condition for Δ0° (**b**) and Δ25° (**c**). Data points averaged across participants were fit using a cumulative normal function. The abscissa refers to 10 orientation differences between the first and second gratings. The ordinate refers to the percentage of trials in which participants indicated the orientation of the second grating that was anticlockwise or clockwise to the first for expected 20° (left) and 70° (right) conditions, respectively. The slope (an index for the Tuning sharpening model), (**d**) and shift (an index for the Tuning shift model), (**e**) differences between the baseline and main experiments for expected 20° and 70° conditions. Statistical comparisons were performed

*Figure 7 continued on next page*

*Figure 7 continued*

using t-tests against zero. Negative: shift to the left; Positive: shift to the right. Open symbols represent the data from each participant and error bars indicate 1 SEM calculated across participants (N=18). B20°: Baseline 20°; B70°: Baseline 70°; E20°: Expect 20°; E70°: Expect 70° (*p<0.05; **p<0.005).

The online version of this article includes the following figure supplement(s) for figure 7:

**Figure supplement 1.** Accuracies of auditory tone reports in orientation discrimination experiments.

and SFD tasks, respectively. For both tasks, the DPCNN was independently and randomly trained 12 times, and for each distance (Δ0°-Δ40°), the training effect was defined as the accuracy difference ($ACC_{difference}$) between the pre- ($ACC_{baseline}$) and post- ($ACC_{trained}$) training. Similar to our psychophysical results, on both tasks, the LR/BFs were much larger than 1 (OD task: LR/BF = $2.045 \times 10^5$, *Figure 8d*, left; SFD task: LR/BF = 5.5929, *Figure 8h*, left) and therefore strongly favored the Mexican-hat model over the Gaussian model. The model comparison based on fitting individual data advocated that the Mexican-hat model was favored over the Gaussian model in 10 and 11 of 12 training data on OD (*Figure 8e*, left) and SFD (*Figure 8i*, left) tasks, respectively. Besides, across individual data, a non-parametric Wilcoxon signed-rank test was conducted to compare the $R^2$ of two models, and results significantly advocated the Mexican-hat model over the Gaussian model on both OD (z=2.197, p=0.028, effect size: r=0.634) and SFD (z=2.981, p=0.003, effect size: r=0.861) tasks. These results suggest that our DPCNN can emerge the similar center-surround inhibition by expectation on both the orientation and spatial frequency trainings.

Additionally, to further determine the contribution of predictive (reconstructive) feedback to center-surround inhibition in expectation, we performed ablation studies, in which we trained the same network but removed feedback predictive coding iterations (a standard feedforward CNN, that is a modified network of AlexNet *Krizhevsky et al., 2012*). As expected, on both tasks, removing feedback leads to the disappearance of center-surround inhibition in expectation. Across individual data, there was no significant difference in the $R^2$ between Mexican-hat and Gaussian models on either OD (non-parametric Wilcoxon signed-rank test: z=–0.314, p=0.754, effect size: r=0.091, *Figure 8e*, right) or SFD (non-parametric Wilcoxon signed-rank test: z=–0.356, p=0.722, effect size: r=0.103, *Figure 8i*, right) tasks. These results further confirm that the predictive coding feedback plays a critical role in producing the center-surround inhibition in expectation.

## Discussion

The present results provide support for an attentional modulation-independent center-surround inhibition profile of expectation and further reveal its underlying neural computations. Specifically, on both OD and SFD tasks, the finest-grained discrimination performance, indexed by the lowest thresholds, of the expected orientation confirmed the previous notion that expectation had a facilitatory effect on various perceptions (*Kok et al., 2012*; *Cheadle et al., 2015*; *Esterman and Yantis, 2010*; *Mareschal et al., 2013*; *McAuley and Kidd, 1998*; *Stein and Peelen, 2015*; *Stocker and Simoncelli, 2006*). Whereas the coarser-grained discrimination performance, indexed by the higher thresholds, of orientations very similar to the expected orientation relative to orientations more distinct from the expected orientation demonstrated a classical inhibitory zone surrounding the focus of expectation (i.e. the center-surround inhibition profile, *Figure 2*). One could argue that this profile was derived from top-down attention rather than expectation. Compared to unexpected gratings with much lower validity, the expected grating with very high validity in our study, presumably, had more degree of top-down attention that has been proven to display the similar center-surround inhibition profile in orientation space by previous studies (*Fang and Liu, 2019*; *Liu et al., 2023*; *Tombu and Tsotsos, 2008*; *Yoo et al., 2018*) and computational models (*Tsotsos et al., 2001*; *Tsotsos et al., 1995*; *Tsotsos et al., 2008*). In other words, our study may not examine a center-surround inhibition profile of expectation, but instead of top-down attention. It is important to note that, in our study, for each grating, participants performed the same discrimination task at threshold, measured by the QUEST staircase procedure (75% correct; *Watson and Pelli, 1983*), which could maximally (although not completely) control the difference in top-down attention among distances. More importantly, our observed center-surround inhibition profile of expectation in orientation space was independent of attentional modulations by the task relevance of orientation (i.e. OD and SFD tasks, *Figure 2*), consistent with previous findings (*Kok et al., 2012*; *Summerfield and de Lange, 2014*; *Rungratsameetaweemana*

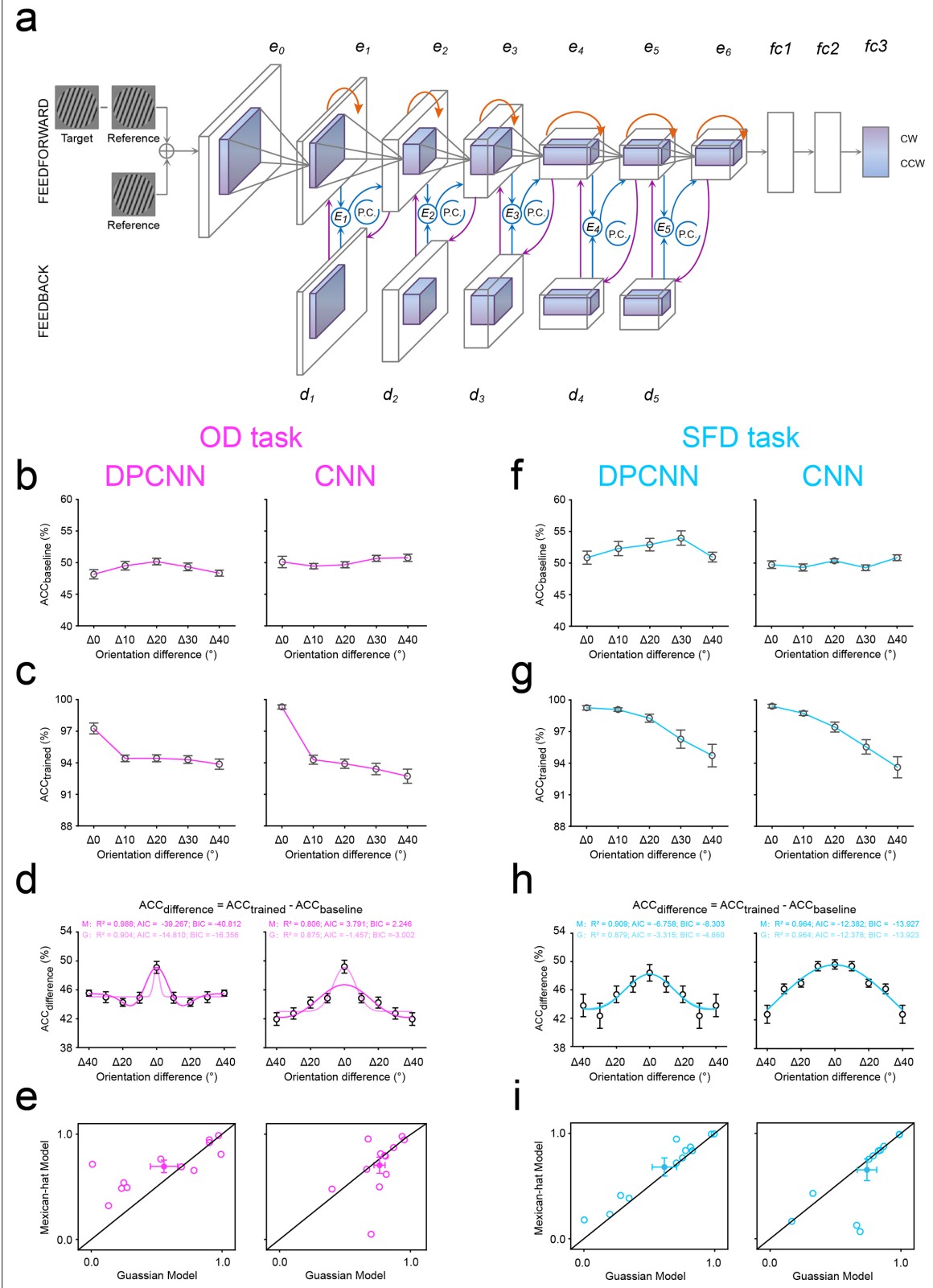

**Figure 8.** Results of artificial neural networks. (**a**) Model structure and stimulus examples for deep predictive coding neural network (DPCNN) and standard feedforward CNN, on both OD (purple) and SFD (blue) tasks. DPCNN consisted of six feedforward encoding layers (e1–e6), five generative feedback decoding layers (d1–d5), and three fully connected (fc) layers. The reconstruction error (E1–E5) is computed and used for the proposed predictive coding updates, denoted by *P.C.* loops. The CNN is the same as DPCNN but removes feedback predictive coding iterations. The accuracy of

*Figure 8 continued*

each distance during the pre- (**b**) and post- (**c**) training for DPCNN (left) and CNN (right), on the OD task. (**d**) The training effect (i.e. the ACC difference between pre- and post-training) of each distance in DPCNN (left) and CNN (right), and the best fitting Mexican-hat and Gaussian functions to these training effects across distances, on the OD task. M, Mexican-hat model; G, Gaussian model. (**e**) $R^2$ of the best fitting Mexican-hat and Gaussian functions from individual data in DPCNN (left) and CNN (right) on the OD task. Open symbols represent individual data and filled colored dots represent the mean across data. Error bars indicate 1 SEM calculated across data (N = 12). (**f–i**) The results from the SFD task, see caption for (**b–e**) for a description of each type of graph.

*and Serences, 2019*; *Summerfield and Egner, 2009*; *Gordon et al., 2019*; *Tal-Perry and Yuval-Greenberg, 2022*; *Wilsch et al., 2020*; *Zuanazzi and Noppeney, 2019*), showing an independency between attention and expectation. If the center-surround inhibition profile was derived from attention rather than expectation, then we should not have observed it on the SFD task, in which the orientation was never task-relevant. Participants did not need to direct attention to this task-irrelevant feature, and therefore yielded none of the profiles in orientation space.

The center-surround inhibition profile of expectation evident in our study is consistent with what has been observed for spatial attention (*Schall and Hanes, 1993*; *Hopf et al., 2006*; *Moran and Desimone, 1985*; *Schall et al., 2004*; *Mounts, 2000*; *Müller and Kleinschmidt, 2004*; *Müller et al., 2005*; *Boehler et al., 2009*; *Boehler et al., 2011*), feature-based attention (*Fang and Liu, 2019*; *Liu et al., 2023*; *Tombu and Tsotsos, 2008*; *Yoo et al., 2018*; *Fang et al., 2019*; *Bartsch et al., 2017*; *Störmer and Alvarez, 2014*; *Loach et al., 2008*), working memory (*Kiyonaga and Egner, 2016*; *Shi et al., 2021*; *Shi et al., 2022*), and visual perceptual learning (*Shen et al., 2024*), in various feature spaces. This suggests that center-surround inhibition could be a unifying principle underlying a diversity of visual representations, as previously proposed by the selective tuning model (*Tsotsos et al., 2001*; *Tsotsos et al., 1995*; *Tsotsos et al., 2008*); however, the extent of the inhibitory zone varied largely across these domains and features. For example, within the orientation space, the inhibitory zone was about 20°, 45°, and 54° for expectation evident here, feature-based attention (*Summerfield et al., 2008*), and visual perceptual learning (*Shen et al., 2024*), respectively; within the feature-based attention, it was about 30° and 45° in color (*Störmer and Alvarez, 2014*) and motion direction (*Fang and Liu, 2019*) spaces, respectively. These variations hint at the exciting possibility that the width of the inhibitory surround may flexibly adapt to stimulus context and task demands, ultimately facilitating our perception and behavior in a changing environment. This principle is consistent with the hybrid model of feature-based attention (*Fang and Liu, 2019*; *Liu et al., 2023*; *Fang et al., 2019*), where attention is deployed adaptively to prioritize task-relevant information through feature-similarity gain which filters out the most distinctive distractors, and surround suppression which inhibits similar and confusable ones, thereby jointly shaping the attentional tuning profile. Mechanistically, the center-surround inhibition profile can be optimal to locally resolve competition between inputs that overlap in their neural representations, specifically attenuating the interference from nearby irrelevant and confusable representations that would be presumably within the same cortical map, and therefore at the largest risk to confuse the current processing. Given the presence of a well-defined map-based organization of the cerebral cortex (*Eickhoff et al., 2018*; *Glasser et al., 2016*; *Mountcastle, 1997*; *Tanaka, 2003*; *Van Essen and Glasser, 2018*; *Wandell et al., 2007*), the center-surround inhibition would be beneficial across all features and therefore serves as a canonical neural computation that sharpens various cognitive processing across different domains.

Strikingly, we found that the center-surround inhibition profile of expectation observed behaviorally can be accounted for by sharpening of tuning curves of neurons of the expected orientation, as revealed by the computational model (*Figure 3*), orientation adjustment (*Figures 5 and 6*), and orientation discrimination (*Figure 7*) experiments. These changes – sharpening of tuning curves – are not only in line with the sharpening hypothesis of expectation developed by previous neurophysiological (*Bell et al., 2016*; *Fiser et al., 2016*; *Kaposvari et al., 2018*; *Meyer and Olson, 2011*; *Schwiedrzik and Freiwald, 2017*), electro-/magneto-encephalogram (*Aitken et al., 2020*; *Kok et al., 2017*; *Todorovic et al., 2011*; *Sedley et al., 2016*; *Wacongne et al., 2011*), and fMRI (*Kok et al., 2012*; *Alink et al., 2010*; *Summerfield et al., 2008*; *Yon et al., 2018*) studies that have invoked the tuning sharpening as the neural basis of expectation-related effects (e.g. the sharpening of tuning curves facilitates fine orientation discrimination by increasing the activity difference between similar orientations), but also extend this hypothesis by identifying the same neural computation in its center-surround

inhibition profile. More importantly, we further found that this profile of expectation can be accounted for by the tuning shift computation that neurons of unexpected orientations shift their spectral tuning toward the expected orientation (*Figures 3 and 6*, and *Figure 7*). We note that our implementation of sharpening and shift at the neuronal level serves as a conceptual model simplification, as population-level tuning, voxel-level selectivity, and behavioral adaptive outcomes may reflect different underlying neuronal mechanisms and do not necessarily align in a one-to-one fashion. Here, we stress that other potential mechanisms beyond sharpening, such as tuning shift, may also contribute to visual expectation. In accordance with expectation, several other cognitive processing tasks have also been shown to shift neuronal tuning curves or receptive fields toward the target, such as spatial (*Connor et al., 1997*; *Fox et al., 2023*; *Klein et al., 2014*; *Sheremata and Silver, 2015*; *Tolias et al., 2001*; *Vo et al., 2017*; *Womelsdorf et al., 2006*) and feature-based (*Çukur et al., 2013*; *Motter, 1994*; *van Es et al., 2018*) attention, as well as visual search (*Tsotsos et al., 1995*; *Carrasco et al., 2004*; *Compte and Wang, 2006*; *Lee et al., 1999*; *Olshausen et al., 1993*; *Rao and Ballard, 1997*) and perceptual learning (*Hanson, 1959*; *Schumacher et al., 2022*; *Spence, 1937*). Interestingly, several brain imaging studies have reported that expectation alters the baseline (*Kok et al., 2017*; *Lucci et al., 2016*; *van Ede et al., 2010*) or gain (*Summerfield and Koechlin, 2008*; *Foley et al., 2017*; *Kok et al., 2016*; *Voss et al., 2008*) of neurons in visual areas, consistent with a classical hypothesis, that is the labeled-line theories of visual information processing (*Adrian and Matthews, 1927*; *Barlow, 1972*; *David et al., 2008*; *Doetsch, 2000*; *Marr, 1982*), which posits that neurons in sensory cortex act as labeled lines with fixed tuning properties that encode input features consistently, regardless of task demands. However, this theory does not account for either tuning curve sharpening or tuning curve shifts of sensory neurons induced by expectation in our study. These changes we observed in the spectral tuning profiles of sensory neurons, conversely, are not only strongly supported by the matched filter hypothesis that neurons could act as matched filters and reshape or shift their tuning to match the target exactly (*David et al., 2008*), but also compatible with both proposals from computational models (*Tsotsos et al., 1995*; *Compte and Wang, 2006*) and Kalman filtering schemes for the signal detection (*Rao and Ballard, 1997*).

Although our study succeeded in linking the center-surround inhibition profile of expectation directly with the response of sensory neurons whose tuning properties make them optimal for demarcating the expected information from various unexpected information, we cannot deny a potential contribution from other cognitive processes, such as decision making. Indeed, previous studies have indicated that expectations primarily influence decisions by modulating post-perceptual stages of information processing (*Summerfield and de Lange, 2014*; *Bang and Rahnev, 2017*; *Gold and Stocker, 2017*; *Rungratsameetaweemana et al., 2018*) or modulate interactions between lower sensory and higher decision areas (*Foley et al., 2017*; *Rahnev et al., 2011*). In addition, these changes in the spectral tuning profiles of sensory neurons evident here derive mainly from psychophysics and computational models. To fully understand how changes in sensory responses contribute to both expectation and its center-surround inhibition profile, further work is needed using neurophysiological techniques or ultra-high field fMRI to explore the locus of events responsible for expectation-induced changes, the identity of neurons that undergo these changes, their patterns of connections, their interactions with higher decision processing, and underlying synaptic bases, especially for our observed shifts in unexpected orientation tunings.

In addition, the emerged center-surround inhibition of expectation in the pretrained DPCNN is not only in line with previous studies and theories that interpret expectation within the predictive coding framework (*Kok et al., 2012*; *Lee and Mumford, 2003*; *Feldman and Friston, 2010*; *Friston, 2005*; *Rao and Ballard, 1999*; *Summerfield and Koechlin, 2008*; *Yuille and Kersten, 2006*), but also adds strong evidence supporting artificial neural networks' potential to perform various human-like representations, such as visual perceptual learning (*Shen et al., 2024*; *Manenti et al., 2023*; *Wenliang and Seitz, 2018*) and hierarchical coding (*Bashivan et al., 2019*), face processing (*Zhou et al., 2022*), contour integration (*Boutin et al., 2021*), and the perception of illusory contours (*Pang et al., 2021*). More importantly, our ablation studies further confirm a critical role of the predictive coding feedback in producing the center-surround inhibition in expectation (*Figure 8*). Although our similarities between artificial neural networks and humans were mostly qualitative, the artificial neural network can provide new ways of studying expectation from behavior to physiology, serving as a test bed for various theories and assisting in generating predictions for physiological studies.

In sum, our study provides, to the best of our knowledge, the first evidence for a center-surround inhibition profile of expectation and how it is supported by not only changes in the tuning curves of sensory neurons but also the predictive coding framework, leading the way towards diversifying models or theories and taking a significant step in unraveling the neuronal computations underlying expectation, or, more generally, top-down processing.

## Methods

### Participants

A total of 24 healthy human adults (16 females, 19–26 years old) were involved in the study. All of them participated in the profile experiments, 20 and 18 of them participated in the orientation adjustment and orientation discrimination experiments, respectively. The sample size was determined based on previous studies investigating visual expectation (*Kok et al., 2012*; *Kok et al., 2017*). All participants had normal or corrected-to-normal vision, were right-handed, and were naive to the purpose of the experiments. They all provided written informed consent for participation and publication. The procedures and protocols were approved by the Human Participants Review Committee of the School of Psychology at South China Normal University and were conducted in accordance with the Declaration of Helsinki.

### Apparatus

The experiments were conducted in a dark, acoustically shielded room. Visual stimuli were displayed on an IIYAMA color graphic monitor (model: HM204DT; refresh rate: 60 Hz; resolution: 1280×1024; size: 22 inches) at a viewing distance of 57 cm. Participants' head position was stabilized using a chin rest.

### Experimental stimuli

Visual stimuli were two consecutive sinusoidal grating stimuli (1.0 contrast, random phase, radius 10°), which were generated using MATLAB (MathWorks, Natick, MA) in conjunction with the Psychophysics Toolbox (*Brainard and Vision, 1997*), and displayed centrally on the gray background (11.196 cd/m²). A white fixation point (radius 0.278°) was always presented at the center of the screen throughout the experiment. The auditory cue consisted of two pure tones (240 Hz and 540 Hz), presented over earphones.

### Experimental design and statistical analysis

#### Profile experiment

##### Experimental design

The profile experiment consisted of baseline and main experiments, and the baseline experiment always preceded the main experiment. The two experiments were the same, except for the predicting probability relationship between the auditory cue and the orientation (20°, 30°, 40°, 50°, 60°, and 70°) of the first grating. For the baseline experiment, the auditory cue, comprising either a low- (240 Hz) or high- (540 Hz) frequency tone, predicted the orientation of the first grating with equal validity (16.67%, *Figure 1a*, left). In the main experiment, this low- or high-frequency tone auditory cue predicted the orientation (20° or 70°) of the first grating with 75% validity. In the remaining 25% of trials, this orientation was chosen randomly and equally from four non-predicted orientations (30°, 40°, 50°, and 60°, *Figure 1a*, right). Thus, for each participant, there were two types of expected conditions: Expect 20° and Expect 70°, and for both conditions, there were five possible distances in orientation space between the expected and test gratings, ranging from Δ0° through Δ40° with a step size of 10°. Note that the matches between the tone (low- or high-frequency) of auditory cue and the expected orientation (20° or 70°) of the first grating were flipped across participants, and the order was also counterbalanced across participants. For each participant, although the tone of auditory cue could not predict 20° or 70° orientation in the baseline experiment, whose trials with the same tone that was matched with 20° or 70° orientation in the main experiment, were defined as Baseline 20° (i.e. the baseline of Expect 20°) and Baseline 70° (i.e. the baseline of Expect 70°) conditions, respectively.

Both the baseline and main experiments consisted of two tasks: the orientation discrimination (OD) task and spatial frequency discrimination (SFD) task, with the two tasks occurring on different days;

the order of the two tasks was counterbalanced across participants. Differently, the baseline experiment consisted of four blocks (two for OD task and the other two for SFD task), and each block had two QUEST staircases (*Watson and Pelli, 1983*) for each of six orientations (20°, 30°, 40°, 50°, 60°, and 70°). The main experiment consisted of two blocks (one for OD task and the other one for SFD task), and each block had 24 QUEST staircases for the expected orientations (20° and 70°) and two QUEST staircases for each of unexpected orientations (30°, 40°, 50°, and 60°). Each QUEST staircase comprised 40 trials, and on each trial, a low- (240 Hz) or a high- (540 Hz) frequency tone (i.e. the auditory cue) was randomly and equally presented for 200 ms, followed by an 1800ms fixation interval. Then, two consecutive gratings were each presented for 150 ms and separated by a 300-ms blank interval (*Figure 1b*). Participants were first asked to make a two-alternative forced-choice (2AFC) judgment of either the orientation (clockwise or anticlockwise, where orientation was task-relevant) or the spatial frequency (lower or higher, where orientation was task-irrelevant) of the second grating relative to the first, on the OD and SFD tasks, respectively. Then, participants were required to make another 2AFC judgment on tone of the auditory cue, either low or high. In the baseline experiment, for both OD and SFD tasks, the orientation of the first grating was chosen randomly and equally from 20°, 30°, 40°, 50°, 60°, and 70°, while its spatial frequency was fixed at 0.9 cycles/°. The second grating differed slightly from the first in terms of both orientation and spatial frequency. Differently, for the OD task, its orientation difference ($\Delta\theta°$, where orientation was task-relevant) varied trial by trial and was controlled by the QUEST staircase to estimate participants' OD thresholds (75% correct), while its spatial frequency difference was set at 0.06 cycle/°; for the SFD task, its spatial frequency difference ($\Delta\lambda$ cycles/°, where orientation was task-irrelevant) varied trial by trial and was controlled by the QUEST staircase to estimate participants' SFD thresholds (75% correct), while its orientation difference was set at 4.8° based on pretest data. Similarly, for each participant, the discrimination threshold obtained during the baseline experiment was used to set the undiscriminated feature difference (i.e. the spatial frequency and orientation for OD and SFD tasks, respectively) during the main experiment, to make the stimuli as similar as possible in both contexts.

## Model fitting and comparison

In both OD and SFD tasks, for two expected conditions and each distance (i.e. Δ0° - Δ40°), we computed a discrimination sensitivity (*DS*) to quantify how much the discrimination threshold (*DT*) changed between baseline (*$DT_{baseline}$*) and main (*$DT_{main}$*) experiments: DS = $DT_{baseline}$ - $DT_{main}$. Because the *DS* from two expected conditions (Expect 20° and Expect 70°) showed a similar pattern, they were pooled together for further analysis (unless otherwise stated, we present average data from two expected conditions). During both tasks, for each participant, a monotonic model and a non-monotonic model to the averaged *DS* were fitted. The monotonic and non-monotonic models were implemented as the Gaussian and Mexican-hat (i.e. a negative second derivative of a Gaussian function) functions (*Shen et al., 2024*; *Wang et al., 2021*), respectively, as follows:

Gaussian function: $y = y0 + \frac{2A}{W\sqrt{2\pi}} e^{-2\left(\frac{x}{w}\right)^2}$

Mexican-hat function: $y = \frac{2H}{\sqrt{3m}\pi^{\frac{1}{4}}} e^{\frac{-x^2}{2m^2}} \left(1 - \frac{x^2}{m^2}\right) + y1$

where *y* is the measured *DS*, *x* is the distance; *w*, *A*, and *y0* are the three parameters controlling the shape of the Gaussian function; *m*, *H*, and *y1* are three parameters controlling the shape of the Mexican-hat function. To compare these two models to our data, we first computed the Akaike information criterion (*AIC*) (*Akaike, 1973*) and Bayesian information criterion (*BIC*) (*Schwarz, 1978*), with the assumption of a normal error distribution as follows:

$$AIC = Nln\left(\frac{RSS}{N}\right) + 2K + \frac{2K\left(K+1\right)}{N - K - 1}$$

$$BIC = Nln\left(\frac{RSS}{N}\right) + Kln\left(N\right)$$

where *N* is the number of observations, *K* is the number of free parameters, and *RSS* is residual sum of squares (*Raftery, 1999*). Then, we further calculated the likelihood ratio (*LR*) and Bayes factor (*BF*) of the non-monotonic models (Mexican-hat) over monotonic model (Gaussian) based on *AIC* (*Burnham and Anderson, 2002*) and *BIC* (*Wagenmakers, 2007*) approximation, respectively, as follows;

$$LR = e^{\left(\frac{AIC_G - AIC_M}{2}\right)}$$

$$BF = e^{\left(\frac{BIC_G - BIC_M}{2}\right)}$$

where $AIC_G$ and $BIC_G$ are for the Gaussian model and $AIC_M$ and $BIC_M$ are for Mexican-hat models.

## Computational models of the center-surround inhibition in expectation

Prior to initiating model fitting, for both OD and SFD tasks, we first transformed the negative values of thresholds during baseline and main experiments into smooth population response profiles using linear interpolation, respectively. Subsequently, we fitted two candidate models, namely Tuning sharpening model and Tuning shift model (*Figure 3a*), to these population response profiles for each participant. In both models, the idealized tuning function for each channel was defined by the Gaussian functions:

$$f(x) = A * e^{\frac{-(x - x0)^2}{\sigma^2}}$$

where *x* is the grating orientation, *A* is the amplitude of tuning function, *x0* is the location, and $\sigma$ is the width. Six and five tuning channels were hypothesized for data in baseline and main experiments, respectively. For the Tuning sharpening model, the tuning width of each channel's tuning function is parameterized by $\sigma$, while all tuning functions are evenly distributed with 10° spacing on the x-axis and the areas under the curves (response energy) are identical. Conversely, for the Tuning shift model, the location of each channel's tuning function is parameterized by *x0*, while they all share the same tuning amplitude and width. The parameter *x0* was constrained within ± 5° of the grating orientation limits, ranging from 15° to 75° during the baseline experiment, 15° to 65° and 25° to 75° for expected 20° and expected 70° conditions, respectively, during the main experiment. The parameter $\sigma$ was set within the range of 0.01–200 to ensure the comparable goodness of fit. For both models, parameters were varied to obtain the minimal sum of squared errors between the population response profile and the model prediction, which is the sum of all channels' tuning responses. To statistically compare the two models, for both orientation and SF discrimination tasks, we computed the root mean squared deviation (RMSD; *Pitt et al., 2002*) of the two fitted models for each participant during baseline and main experiments:

$$\text{RMSD} = \sqrt{\frac{SSE}{(N - K)}}$$

where *SSE* is the sum of squared errors. *N* is the number of data points (i.e. 51 and 61), and *K* is the number of model parameters.

## Orientation adjustment experiment

### Experimental design

The protocol of orientation adjustment experiment was similar to that of the profile experiment, except for two aspects. First, there were four possible (20°, 40°, 50°, and 70°) orientations for the first grating: 20°/70° (Δ0° deviated from the expected orientation) and 40°/50° (Δ20°/Δ30° deviated from the expected orientation). Second, in both baseline and main experiments, the second grating was set as a random orientation within the range of 0° to 90°, and participants were required to rotate the orientation of the second grating to match the first (*Figure 5a*). Each participant completed 8 blocks of 48 trials in the baseline experiment and 16 blocks of 48 trials in the main experiment.

### Modeling response error

Response error was measured as the angular difference between the orientation of the first grating and the adjusted orientation of the second grating, such that errors ranged from 0° (a perfect response) to ± 90° (a maximally imprecise response). To evaluate performance, we categorized the response errors for each participant according to different conditions and modeled their distributions

as a three-component mixture model (*Suchow et al., 2013*). This model comprised a von Mises distribution ($\varnothing$) corresponding to trials in which the grating orientation was encoded and a uniform distribution ($p_g$) accounting for the probability of random guessing without encoding (*Zhang and Luck, 2008*):

$$p\left(\theta\right) = \left(1 - p_g\right)\varnothing_{\mu,k} + p_g\left(\frac{1}{2\pi}\right)$$

where $\theta$ is the adjusted orientation value, $\varnothing$ denotes the Von Mises distribution with mean $\mu$ and shape parameter $k$, and $p_g$ represents a uniform distribution. Specifically, the von Mises probability density function for the angle $x$ is given by:

$$\varnothing\left(x \vee \mu, k\right) = \frac{e^{kcos\left(x-\mu\right)}}{2\pi I_0\left(k\right)}$$

where $I_0\left(k\right)$ is the modified Bessel function of the first kind of order 0, with this scaling constant chosen so that the distribution sums to unity:

$\int_{-\pi}^{\pi} e^{kcosx}\,dx = 2\pi I_0\left(k\right)$

Here, we obtained maximum likelihood estimates for 3 parameters: (1) the systematic shift of von Mises distribution (*mu*), which reflects distribution shift away from the target grating orientation; (2) the dispersion of the von Mises distribution (*s.d.*=$\sqrt{1/k}$), which reflects response precision or resolution of representation; and (3) the height of the uniform distribution (*g*), which reflects the probability of guessing.

## Orientation discrimination experiment
### Experimental design
The protocol of orientation discrimination experiment was very similar to that of the orientation adjustment experiment, except for two aspects (*Figure 7a*). First, there were three possible (20°, 45°, and 70°) orientations for the first grating: 20°/70° (Δ0° deviated from the expected orientation) and 45° (Δ25° deviated from the expected orientation). Second, in both baseline and main experiments, the second grating was 1°, 3°, 5°, 7°, and 9° deviated from the first grating, either clockwise or counterclockwise. Participants were asked to make a 2AFC judgment of the orientation of the second grating relative to the first, either clockwise or anticlockwise. Each participant completed 10 blocks of 120 trials in the baseline experiment, and 20 blocks of 160 trials in the main experiment.

## Data fitting and analysis
We first constructed a psychometric function for each condition shown in *Figure 7*. We plotted the percentage of trials in which participants indicated the orientation of the second grating that was anticlockwise or clockwise to the first for 20° (Baseline 20° and Expect 20°) and 70° (Baseline 70° and Expect 70°) conditions, respectively, as a function of the real orientation difference between two gratings. For each participant and each condition, the psychometric values at ten orientation differences were fitted to a cumulative Gaussian using Bayesian inference, implemented in the *Psignifit toolbox* for Matlab (Version 4; *Schütt et al., 2016*), and we interpolated the data to find the slope (orientation uncertainty) and PSE (point of subjective equality, which is the shift here) as an index for Tuning sharpening and Tuning shift models, respectively.

## Artificial neural networks for the center-surround inhibition in expectation
We trained a deep predictive coding neural network (DPCNN), modified from Predify (*Choksi and Mozafari, 2021*; *Pang et al., 2021*) to perform both the OD and SFD tasks. Relative to the reference, on the OD task, DPCNN was trained to classify whether the target was tilted clockwise or counterclockwise; whereas on the SFD task, it was trained to classify whether the target had lower or higher spatial frequency. For both tasks, the DPCNN consisted of six feedforward encoding layers (*e1-e6*), five generative feedback decoding layers (*d1-d5*), and three fully connected (*fc*) layers (*Figure 8a*).

The reconstruction error (*E1-E5*) is computed and used for the proposed predictive coding updates (**Rao and Ballard, 1999**), denoted by *P.C.* loops. Note that the updating is only applied to *e1-e5*, and for the last layer *e6*, there is no feedback. Before the layer *e0*, we obtained the pixel difference between the target and reference images, which was then superimposed on the channels of the reference image. This superimposed image was set as the input of the network and remained constant over timesteps. Besides, we used a feedforward encoding layer (i.e. *e0*) to match the number of the channels between superimposed feature maps and the pre-trained DPCNN. Additionally, to further determine the contribution of predictive coding framework to center-surround inhibition in expectation, we also trained the same network but removed feedback predictive coding iterations (a standard feedforward CNN, i.e. a modified network of AlexNet **Krizhevsky et al., 2012**). Note that all these architects were built to mimic our hypothesis of the visual pathway involved in expectation (**de Lange et al., 2018**; **Press et al., 2020**; **Kok et al., 2012**; **Summerfield and de Lange, 2014**; **Summerfield and Egner, 2009**) and could learn a lower-dimensional latent representation of a high-dimensional input space (**Kingma and Welling, 2013**), similar to prior-based low-light image enhancement (**Wu et al., 2025**). During the training, the last layer was trained to capture the difference between the target and reference and finally obtain the classification by softmax, to model decision making in our 2AFC paradigm (**Figure 1c**), in which participants were asked to make a 2AFC judgment of the orientation (either clockwise or anticlockwise) or the spatial frequency (either lower or higher) of the second grating (target) relative to the first (reference) in OD and SFD tasks, respectively. Moreover, for the OD task, the orientation difference between the target and reference in the network was set to 5°; for the SFD task, the spatial wavelength difference between them was set to 0.5.

For each task, both the DPCNN and CNN were independently and randomly trained 12 times. For each time, the trained orientation was chosen randomly from 0° to 180°; the 9 test gratings were 0°, ±10°, ±20°, ±30°, and ± 40° deviated (clockwise and counterclockwise) from the trained orientation. All grating stimuli (phase: random) were centered on 227×227-pixel images with gray background. To improve the robustness of our model, we trained the network on all combinations of several parameters: contrast (0.1, 0.15, 0.2, 0.25, 0.3, 0.4, 0.5, 0.6, 0.7, and 0.8), SD of the Gaussian additive noise (5, 25, and 45), and spatial wavelength (5, 10, 15, 20, 25, 30, 40, 50, 60, and 80 pixels) for the OD task; contrast ranging from 0.1 through 0.8 with a step size of 0.05 and SD of the Gaussian additive noise ranging from 3 through 60 with a step size of 3 for the SFD task. For each training, there were thus a total of 840 images; 600 images were the training set and the other 240 images were the test set. For both OD and SFD tasks, during the training set, there were 480 images for the expected orientation (Δ0°) and 30 images for each of unexpected orientations (Δ10°, Δ20°, Δ30°, and Δ40°); during the test set, there were 48 images for each of distances (Δ0°- Δ40°). For each distance, the training effect was defined as the accuracy difference ($ACC_{difference}$) between the pre- ($ACC_{baseline}$) and post- ($ACC_{trained}$) training.

## Acknowledgements

We acknowledge the participants for their contribution to this study. XZ was supported by the National Natural Science Foundation of China (32271099), the Research Center for Brain Cognition and Human Development of Guangdong Province (2024B0303390003), and the Striving for the First-Class, Improving Weak Links and Highlighting Features (SIH) Key Discipline for Psychology in South China Normal University. R-YZ was supported by the National Natural Science Foundation of China (32441102) and the Shanghai Municipal Education Commission (2024AIZD014).

## Additional information

### Competing interests

Xilin Zhang: Reviewing editor, eLife. The other authors declare that no competing interests exist.

## Funding

| Funder | Grant reference number | Author |
| --- | --- | --- |
| National Natural Science Foundation of China | 32271099 | Xilin Zhang |
| Research Center for Brain Cognition and Human Development of Guangdong Province | 2024B0303390003 | Xilin Zhang |
| Striving for the First-Class, Improving Weak Links and Highlighting Features (SIH) Key Discipline for Psychology in South China Normal University | | Xilin Zhang |
| National Natural Science Foundation of China | 32441102 | Ru-Yuan Zhang |
| Shanghai Municipal Education Commission | 2024AIZD014 | Ru-Yuan Zhang |

The funders had no role in study design, data collection and interpretation, or the decision to submit the work for publication.

## Author contributions

Ling Huang, Data curation, Formal analysis, Investigation, Visualization, Methodology; Shiqi Shen, Data curation, Formal analysis, Investigation, Methodology, Writing – review and editing; Yueling Sun, Formal analysis, Investigation, Methodology, Writing – review and editing; Shipei Ou, Investigation; Ru-Yuan Zhang, Funding acquisition, Methodology; Floris P de Lange, Methodology, Writing – review and editing; Xilin Zhang, Conceptualization, Supervision, Funding acquisition, Writing – original draft, Project administration, Writing – review and editing

## Author ORCIDs

Ling Huang ⓘ https://orcid.org/0009-0006-9712-2726
Shiqi Shen ⓘ https://orcid.org/0000-0001-9082-9178
Ru-Yuan Zhang ⓘ https://orcid.org/0000-0002-0654-715X
Floris P de Lange ⓘ https://orcid.org/0000-0002-6730-1452
Xilin Zhang ⓘ https://orcid.org/0000-0003-0449-934X

## Ethics

A total of 24 healthy human adults (16 females, 19-26 years old) were involved in the study. All of them participated in the profile experiments, 20 and 18 of them participated in the orientation adjustment and orientation discrimination experiments, respectively. The sample size was determined based on previous studies investigating visual expectation. All participants had normal or corrected-to-normal vision, were right-handed, and were naïve to the purpose of the experiments. They all provided written informed consent for participation and publication. The procedures and protocols were approved by the Human Participants Review Committee of the School of Psychology at South China Normal University and were conducted in accordance with the Declaration of Helsinki.

Reviewer #1 (Public review): https://doi.org/10.7554/eLife.107301.3.sa1
Reviewer #2 (Public review): https://doi.org/10.7554/eLife.107301.3.sa2
Author response https://doi.org/10.7554/eLife.107301.3.sa3

---

# Additional files

## Supplementary files

Supplementary file 1. Supplementary materials containing all additional data, analyses, and supporting results for the study.

MDAR checklist

## Data availability

The datasets and codes for this study are available at Open Science Framework https://osf.io/5tj8c/.

The following dataset was generated:

| Author(s) | Year | Dataset title | Dataset URL | Database and Identifier |
|---|---|---|---|---|
| Zhang X | 2024 | Center-surround inhibition in expectation and its underlying computational and artificial neural net | https://osf.io/5tj8c | Open Science Framework, 5tj8c |

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
