## [Editor Report · eLife Assessment]

This is a methodologically rich manuscript that is **important** for revealing the center-surround inhibition profile of expectation in orientation space. The analyses are **compelling** in validating the critical role of predictive coding feedback. The findings provide novel insights into how expectation optimizes perception via enhancement and suppression.

---

## [Referee Report · Reviewer #1 (Public review)]

Summary:

The authors tested two competing mechanisms of expectation (1) a sharpening model that suppresses unexpected information via center-surround inhibition; (2) a cancellation model that predicts a monotonic gradient response profile. Using two psychophysical experiments manipulating feature space distance between expected and unexpected stimuli, the results consistently supported the sharpening model. Computational modeling further showed that expectation effects were explained by either sharpened tuning curves or tuning shifts. Finally, convolutional neural network simulations revealed that feedback connections critically mediate the observed center-surround inhibition.

Strengths:

The manuscript provides compelling and convergent evidence from both psychophysical experiments and computational modeling to robustly support the sharpening model of expectation, demonstrating clear center-surround inhibition of unexpected information.

Comments on revisions:

I appreciate the authors' thoughtful revisions. I have no further comments.

---

## [Referee Report · Reviewer #2 (Public review)]

Summary:

This is a compelling and methodologically rich manuscript. The authors used a variety of methods, including psychophysics, computational modeling, and artificial neural networks, to reveal a non-monotonic, center-surround "Mexican-hat" profile of expectation in orientation space. Their data convincingly extend analogous findings in attention and working memory, and the modeling nicely teases apart sharpening vs. shift mechanisms.

Strengths:

The findings are novel and important in elucidating the potential neural mechanisms by which expectation shapes perception. The authors conducted a series of well-designed psychophysical experiments to careful examination of the profile of expectation's modulation. Computational modeling also provides further insights, linking the neural mechanisms of expectation to behavioral results.

Comments on revisions:

I think the authors did a great job in addressing my previous comments. I have no further comments.

---

## [Author Response]

The following is the authors’ response to the original reviews.

**Reviewer #2 (Public review):**
(1) The sharpening model of expectation can predict surround suppression. The authors could further clarify how the cancellation model predicts a monotonic profile of expectation (Figure 1C) with the highest response at the expected orientation, while the cancellation model suggests a suppression of neurons tuned toward the expected stimulus.

We thank the reviewer for the comment. We would like to emphasize that as the expected signal is suppressed, the relative weight or salience of unexpected inputs increases. We have clarified this interpretation in the manuscript as follows:

“Here, given these two mechanisms making opposite predictions about how expectation changes the neural responses of unexpected stimuli, thereby displaying different profiles of expectation, we speculated that if expectation operates by the sharpening model with suppressing unexpected information, we should observe an inhibitory zone surrounding the focus of expectation, and its profile then should display as a center-surround inhibition (Fig. 1c, left). If, however, expectation operates as suggested by the cancelation model with highlighting unexpected information, the inhibitory zone surrounding the focus of expectation should be eliminated, and the profile should instead display a monotonic gradient (Fig. 1c, right).”

(2) I'm a bit concerned about whether the profile solely arises from modulation of expectation. The two auditory cues are each associated with a fixed orientation, which may be confounded by other cognitive processes like visual working memory or attention (which I think the authors also discussed). Although the authors tried to use SFD task to render orientation task-irrelevant, luminance edges (i.e., orientation) and spatial frequency in gratings are highly intertwined and orientation of the gratings may help recall the first grating's SF (fixed at 0.9 c/{degree sign}), especially given the first and second grating's orientations are not very different (4.8{degree sign}).

We agree that dissociating expectation from attention and other top-down processes remains a key challenge in visual expectation research (see Summerfield & Egner, 2009; Summerfield & de Lange, 2014; de Lange et al., 2018). As is generally acknowledged, expectation reflects the probability of a sensory event, while selective attention relates to its behavioral relevance. To minimize attentional influences, our task design ensured that grating orientation was not taskrelevant: on each trial, participants discriminated either orientation or spatial frequency difference, such that orientation itself did not require attentional allocation, a point already discussed in the manuscript.

Regarding visual working memory, we argue that even if participants recalled the first grating’s spatial frequency in the SFD task, they were not required to retain its precise spatial frequency (or orientation), as their task was simply to judge whether the second grating appeared denser or sparser. In other words, orientation (or spatial frequency) itself was not task-relevant. Moreover, although not included in the manuscript, we conducted a post-experiment debriefing in which participants were asked whether they noticed any association between the auditory tone and the grating orientation. None of the participants reported this relationship correctly, suggesting that the tone-orientation mapping remained implicit and was unlikely to be driven by strategic attention or memory.

However, we acknowledge that certain confounding processes such as statistical learning or implicit mapping acquisition cannot be fully ruled out given the current paradigm. Future studies using methods with higher temporal resolution (e.g., EEG/MEG) may help to dissociate these mechanisms more precisely.

(3) For each of the expected orientations (20{degree sign} or 70{degree sign}), the unexpected ones are linearly separable (i.e., all unexpected ones lie on one side of the expected angle). This might further encourage people to shift their attended or expected orientation, according to the optimal tuning hypothesis. Would this provide an alternative explanation to the tuning shift that the authors found?

We thank the reviewer for pointing out the relevance of the optimal tuning hypothesis. We acknowledge that the optimal tuning theory (Navalpakkam & Itti, 2007) is an important framework, particularly in visual search paradigms, where attentional templates may shift away from non-target features to enhance discriminability.

In our task, this hypothesis would predict a shift of expectation toward <20° in E20° trials and >70° in E70° trials, given that all unexpected orientations lie on one side of the expected angle. Importantly, the optimal tuning hypothesis predicts such shifts not only in Δ20°, Δ25°, and Δ30° trials but also in the Δ0° trials. In this regard, the observed shift in Δ20° and Δ30° (Experiment 2) and Δ25° (Experiment 3) trials is broadly consistent with the predictions of the optimal tuning account. However, we did not observe a corresponding shift away from nontarget features in the Δ0° condition, suggesting limited behavioral evidence for optimal tuning effects under our current task settings.

It is important to note that most previous studies supporting optimal tuning (e.g., Navalpakkam & Itti, 2007; Scolari & Serences, 2009; Geng, DiQuattro, & Helm, 2017; Yu & Geng, 2019) have used visual search paradigms that differ from our design in several critical ways, including the number of stimuli presented, their spatial arrangement (eccentricity), task demands, and so on. Therefore, it is difficult to determine whether the optimal tuning hypothesis could serve as an alternative explanation within the context of our current study. We agree that future studies could further examine how such task parameters influence the presence or absence of optimal tuning.

(4) It is great that the authors conducted computational modeling to elucidate the potential neuronal mechanisms of expectation. But I think the sharpening hypothesis (e.g., reviewed in de Lange, Heilbron & Kok, 2018) focuses on the neural population level, i.e., narrowing of population tuning profile, while the authors conducted the sharpening at the neuronal tuning level. However, the sharpening of population does not necessarily rely on the sharpening of individual neuronal tuning. For example, neuronal gain modulation can also account for such population sharpening. I think similar logic applies to the orientation adjustment experiment. The behavioral level shift does not necessarily suggest a similar shift at the neuronal level. I would recommend that the authors comment on this.

We thank the reviewer for this to-the-point comment. As de Lange et al. (2018) noted, “there is not always a direct correspondence between neural-level and voxel-level selectivity patterns.” That is, neuronal tuning, population-level tuning, voxel-level selectivity, and behavioral adaptive outcomes may reflect different underlying mechanisms and do not necessarily align in a one-toone fashion. We fully acknowledge that population-level tuning effects may also result from various neuronal mechanisms such as gain modulation (for review, see Salinas & Thier, 2000), shifts in preferred orientation (Ringach, et al., 1997; Jeyabalaratnam et al., 2013), asymmetric broadening of tuning curves (Schumacher et al., 2022), or tuning curve sharpening (Ringach, et al., 1997; Schoups et al., 2001).

In our modeling, we implemented sharpening and shifts of neuronal tuning curves as a conceptual model simplification, intended to explore potential mechanisms underlying expectation-related center-surround suppression effects. While sharpening-based accounts (e.g., Kok et al. 2012) have often been emphasized, we stress that other mechanisms, such as gain modulation or tuning shifts, may also contribute. Our goal is not to provide a definitive account, but to highlight such plausible mechanisms and encourage future investigation. We have revised the Discussion to emphasize that multiple mechanisms may underlie the observed effects.

“We note that our implementation of sharpening and shifts at the neuronal level serves as a conceptual model simplification, as population-level tuning, voxel-level selectivity, and behavioral adaptive outcomes may reflect different underlying neuronal mechanisms and do not necessarily align in a one-to-one fashion. Here, we stress that other potential mechanisms beyond sharpening, such as tuning shifts, may also contribute to visual expectation.”

(5) If the orientation adjustment experiment suggests that both sharpening and shifting are present at the same time, have the authors tried combining both in their computational model?

We agree with the reviewer that it is necessary to consider the combined model. Accordingly, we implemented a computational model incorporating sharpening of the expected orientation channel together with shifting of the unexpected orientation channels. This model

successfully captured the sharpening of the expected-orientation channel and the shift of the unexpectedorientation channels (Supplementary Fig. 3). For the expected orientation (Δ0°) , results showed that the amplitude change was significantly higher than zero on both OD (t(23) = 2.582, p = 0.017, Cohen’s d = 0.527) and SFD (t(23) = 2.078, p = 0.049, Cohen’s d = 0.424) tasks (Supplementary Fig. 3e, vertical stripes); the width change was significantly lower than zero on both OD (t(23) = -2.438, p = 0.023, Cohen’s d = 0.498) and SFD (t(23) = -2.578, p = 0.017, Cohen’s d = 0.526) tasks (Supplementary Fig. 3e, diagonal stripes). For unexpected orientations (Δ10°-Δ40°), however, the amplitude and width changes were not significant with zero on either OD (amplitude change: t(23) = 0.443, p = 0.662, Cohen’s d = 0.091; width change: t(23) = -1.819, p = 0.082, Cohen’s d = 0.371) or SFD (amplitude change: t(23) = 1.130, p = 0.270, Cohen’s d = 0.231; width change: t(23) = -1.710, p = 0.101, Cohen’s d = 0.349) tasks (Supplementary Fig. 3f). In the meantime, the location shift was significantly different than zero for unexpected orientations (Δ10°-Δ40°), OD task: t(23) = 3.611, p = 0.001, Cohen’s d = 0.737; SFD task: t(23) = 2.418, p = 0.024, Cohen’s d = 0.493 (Supplementary Fig. 3g). These results provided further evidence that tuning sharpening and tuning shift jointly contribute to center– surround inhibition in expectation.

**Reviewer#1 (Recommendation for the Author):**
(1) A direct comparison between tasks (baseline vs. expectation conditions) would have strengthened the findings. Specifically, contrasting performance in the orientation discrimination task with the spatial frequency discrimination task could have provided clearer evidence that participants actually used the auditory cues to attend to the expected orientation. This comparison would be particularly important for validating cue manipulation in the orientation discrimination task.

We agree that a direct comparison between the orientation discrimination (OD) and spatial frequency discrimination (SFD) tasks could further clarify how expectation (auditory cues) differentially modulates orientation relevance. However, the primary goal of the current study was to examine expectation effects within each task separately and to demonstrate that such effects are independent of attentional modulation driven by the task-relevance of orientation.

In addition, the OD and SFD tasks differ not only in the relevant task features (orientation vs. spatial frequency discrimination), but also in stimulus properties and difficulty, for example, the arbitrary use of 20–70° as the orientation range and ~0.9 cycles/° as the spatial frequency setting, a direct comparison could introduce confounding factors unrelated to expectation.

Importantly, Previous studies (e.g., Kok et al., 2012, 2017; Aitken et al., 2020) and our current results show that participants performed significantly better when the auditory cue matched the expected orientation, supporting the validity of our expectation manipulation.

(2) An interesting consideration is why the center-surround inhibition profile of expectation was independent of the task-relevance of orientation. Previous studies (e.g., Kok et al., 2012) have found that orientation discrimination patterns differ depending on whether orientation is taskrelevant or irrelevant. This could be useful to discuss the possible discrepancies.

We thank the reviewer for this inspiring comment. Kok et al. (2012) showed that both orientation and contrast tasks elicited similar fMRI decoding results, regardless of task relevance, suggesting neural mechanisms of expectation operate independently of whether orientation is task relevant. Behaviorally, they reported better performance for expected versus unexpected trials in the orientation task (3.4° vs. 3.8°, t(17) = 2.8, p = 0.013), and a marginal trend (although not significant) in the contrast task (4.3% vs. 5.0%, t(17) = 1.9, p = 0.075). If any differences between the two tasks exist, they may lie in the correlation between behavioral and fMRI effects, a question that goes beyond the scope of the current study. Therefore, it is hard to strongly conclude that orientation discrimination patterns differ depending on whether orientation is taskrelevant or irrelevant in their paper.

Our study differs from theirs in at least two important ways, which may account for the clearer expectation facilitatory effect we observed in the expectation (Δ0°) condition. First, in our study, the orientation-irrelevant task involved spatial frequency discrimination (SFD) rather than contrast discrimination. Compared to contrast, spatial frequency has been shown to exhibit a clear cueing effect, as reported in Fang & Liu (2019). Second, our design included a baseline condition, which was absent in their study. We computed discrimination sensitivity (DS) to quantify how much the discrimination threshold (DT) changed relative to baseline. By using this baseline-referenced approach, we observed a significant facilitatory expectation effect in the Δ0° condition, an effect that shifted from marginal significance in their orientation-irrelevant task to clear significance in our study.

(3) The authors might consider briefly explaining how the orientation adjustment paradigm used in this study is particularly effective for examining the potential co-existence of tuning sharpening and tuning shift computations, and how this approach complements traditional orientation discrimination tasks in characterizing expectation-related mechanisms.

We thank the reviewer for this valuable suggestion. We agree that further clarification is needed to better connect the two experiments. To explain this, we have elaborated further in the manuscript.

“To further explore the co-existence of both Tuning sharpening and Tuning shift computations in center-surround inhibition profile of expectation, participants were asked to perform a classic orientation adjustment experiment. Unlike profile experiment (discrimination tasks), the adjustment experiment provides a direct, trial-by-trial measure of participants’ perceived orientation, capturing the full distribution of responses. This enables the construction of orientation-specific tuning curves, allowing us to detect both tuning sharpening and tuning shifts, thereby offering a more nuanced understanding of the computational mechanisms underlying expectation.”

(4) These interesting findings raise important questions about their relationship to existing hybrid models of attentional modulation. Could the authors discuss how their results might align with or extend previous work demonstrating combined feature-similarity gain and surround suppression effects for orientation (e.g., Fang & Liu, 2019)? Could a hybrid model potentially provide a better account of these data than the pure surround suppression model?

We thank the reviewer for this valuable comment. We agree that hybrid model should be mentioned in the manuscript and we have elaborated further in the Discussion.

“For example, within the orientation space, the inhibitory zone was about 20°, 45°, and 54° for expectation evident here, feature-based attention[21], and visual perceptual learning[35], respectively; within the feature-based attention, it was about 30° and 45° in color [77] and motion direction [53] spaces, respectively These variations hint at the exciting possibility that the width of the inhibitory surround may flexibly adapt to stimulus context and task demands, ultimately facilitating our perception and behavior in a changing environment. This principle is consistent with the hybrid model of feature-based attention [53,54,75], where attention is deployed adaptively to prioritize task-relevant information through feature-similarity gain which filters out the most distinctive distractors, and surround suppression which inhibits similar and confusable ones, thereby jointly shaping the attentional tuning profile.”

(5) On page 19, there appears to be a missing symbol in the description of the Tuning Sharpening model. The text states: 'the tuning width of each channel's tuning function is parameterized by ??', where the question marks seem to indicate a missing parameter symbol.

We appreciate the reviewer’s careful attention. Yes, the "ơ" is missing, which was likely caused by a formatting issue. We have corrected it.